# Functional Surface Coatings on Orthodontic Appliances: Reviews of Friction Reduction, Antibacterial Properties, and Corrosion Resistance

**DOI:** 10.3390/ijms24086919

**Published:** 2023-04-07

**Authors:** Ruichu Zhang, Bing Han, Xiaomo Liu

**Affiliations:** 1Department of Orthodontics, School and Hospital of Stomatology, Peking University, Beijing 100081, China; 2National Center of Stomatology & National Clinical Research Center for Oral Diseases & National Engineering Laboratory for Digital and Material Technology of Stomatology & Beijing Key Laboratory for Digital Stomatology & Research Center of Engineering and Technology for Computerized Dentistry Ministry of Health & NMPA Key Laboratory for Dental Materials, Beijing 100081, China

**Keywords:** coated materials, corrosion, friction, microorganisms, orthodontic brackets, orthodontic wires, orthodontics

## Abstract

Surface coating technology is an important way to improve the properties of orthodontic appliances, allowing for reduced friction, antibacterial properties, and enhanced corrosion resistance. It improves treatment efficiency, reduces side effects, and increases the safety and durability of orthodontic appliances. Existing functional coatings are prepared with suitable additional layers on the surface of the substrate to achieve the abovementioned modifications, and commonly used materials mainly include metal and metallic compound materials, carbon-based materials, polymers, and bioactive materials. In addition to single-use materials, metal-metal or metal-nonmetal materials can be combined. Methods of coating preparation include, but are not limited to, physical vapor deposition (PVD), chemical deposition, sol-gel dip coating, etc., with a variety of different conditions for preparing the coatings. In the reviewed studies, a wide variety of surface coatings were found to be effective. However, the present coating materials have not yet achieved a perfect combination of these three functions, and their safety and durability need further verification. This paper reviews and summarizes the effectiveness, advantages and disadvantages, and clinical perspectives of different coating materials for orthodontic appliances in terms of friction reduction, antibacterial properties, and enhanced corrosion resistance, and discusses more possibilities for follow-up studies as well as for clinical applications in detail.

## 1. Introduction

Surface coating technology relies on various metals, polymers, and composite materials by preparing a suitable additional layer on the surface of the substrate and applying different coatings to the surface of the brackets and archwires to modify their surface morphology, mechanical properties, and antibacterial properties [1]. This is currently an important way to improve the performance of orthodontic appliances, enabling clinical effects such as friction reduction, antibacterial effects, and corrosion resistance.

It is necessary to reduce the friction between the archwire and brackets, improve the antibacterial properties of orthodontic appliances, and increase their corrosion and wear resistance using surface coatings, in order to improve treatment efficiency, reduce the consumption of anchorage, reduce the incidence of caries and periodontal disease caused by orthodontics, and increase the safety and durability of orthodontic attachments. Some studies have shown that friction can offset 12–60% of the orthodontic force during orthodontic treatment [2], reducing the efficiency of tooth movement and increasing the consumption of anchorage, which is detrimental to the treatment [2,3]. Moreover, a white spot lesion (WSL) is often observed on the enamel surface around orthodontic attachments [4,5,6], and plaque-induced gingivitis [7] appears occasionally. These are due to difficulties in the mechanical removal of plaque and increased plaque retention [8]. Furthermore, metal orthodontic appliances are susceptible to corrosion in the intraoral environment [9,10,11]. In addition to electrochemical corrosion, chewing, brushing teeth, and archwires sliding along the brackets, the mechanical forces on the surfaces also accelerate the process of corrosion [10,12], causing the release of various metal ions—especially nickel (Ni) ions [13,14,15]. These metal ions act as allergens, increasing the likelihood of metal allergy for orthodontic patients [16,17]. Allergic reaction of the oral mucosa to nickel can cause distress and pain to patients; this is significantly more common in females than in males, especially between the ages of 16 and 35 years. The discomfort to patients includes burning sensation of the oral mucosa, cheilitis, parageusia, periodontitis, and even erythema multiforme and popular perioral rash [17]. These conditions can reduce patients’ quality of life and oral health, which can adversely affect treatment compliance and therapeutic effects. Thus, surface coatings are important to be put into clinical use.

However, the existing coating materials and methods for orthodontic fixed appliances are numerous and complicated, while the purpose and focus point of the coatings differ. Based on the above considerations, this article mainly reviews and summarizes the effects, advantages, disadvantages, and clinical prospects of different coating materials with respect to friction reduction, antibacterial effects, and corrosion resistance, in order to guide the selection and preparation of suitable surface coatings for clinical applications. Coatings on brackets and archwires can be seen in Figure 1.

## 2. Friction Reduction Coatings

Existing orthodontic friction reduction coatings can be divided into two major categories—metallic and non-metallic—and each major category can be divided into various single coatings and composite coatings. Among metal and metallic compound friction reduction coatings, tungsten disulfide (WS_2_) was the first to be applied [22]. Since then, silver (Ag) coatings and metallic compound coatings, such as zinc oxide (ZnO) [23,24,25,26,27], titanium nitride (TiN) [28,29,30,31,32,33], and aluminum oxide (Al_2_O_3_) [30], have also received good attention. Nonmetals, including carbon-based materials [34], polymers [35,36,37,38,39], and bioactive materials [40], are relatively new coating materials. In addition, preparing nanoparticles and depositing them on orthodontic attachment surfaces via magnetron sputtering [30,33], evaporation [41], and immersion are also common techniques. These materials play an antifriction role mainly by filling the grooves on the surface of the appliances and forming a lubricating layer, which effectively reduces the dynamic and static friction between the archwire and the brackets [42]. The main friction reduction coating materials discussed in this section are shown in Figure 2.

### 2.1. Inorganic Fullerene-like Nanoparticles of Tungsten Disulfide Coatings

A. Katz et al. used nickel-phosphorus (Ni-P) electroless films impregnated with inorganic fullerene-like nanoparticles of tungsten disulfide (IF-WS_2_) to coat stainless steel (SS) archwires and showed that the IF-WS_2_ coating significantly reduced archwire friction and mitigated adverse complications [22]. Similarly, the results of M. Redlich et al. showed a 54% reduction in the friction of the coated archwire compared to the uncoated SS archwires [42]. However, despite the biocompatibility of IF-WS_2_, its cytotoxicity has not been studied in detail [42], which may have a negative impact on the organization. In 2019, a study applied molybdenum disulfide (MoS_2_) and tungsten disulfide (WS_2_) coatings to orthodontic SS archwires via electrodeposition and found that both materials reduced the friction between the archwires and the brackets under dry conditions [18]. However, when fixed appliances are placed inside the oral cavity, the environment to which they are exposed is wet. The wettability and durability of WS_2_ and MoS_2_ has been discussed in other fields [43,44,45], which needs to be taken into account when considering their application in the dental field. Therefore, the ability of MoS_2_ and WS_2_ coatings to reduce friction on archwires and brackets in wet environments—for instance, when immersed in artificial saliva or applied in animals’ oral cavities—needs to be investigated further.

WS_2_ nanoparticles can form a homogeneous and thickness-appropriate coating on the surface of the archwires and brackets; they can also have a similar effect in combination with Ni and Ni-P [22]. The film of nanoparticles peels off when exposed to friction, forming a solid lubricant film between the interfaces, thereby reducing the friction—especially when the pressure is high [42]. Therefore, WS_2_ coatings are more suitable apply during the alignment process where the contact angle is large, i.e., the tipping and uprighting type of tooth movement [22]. Still, it is worth noting that large-angle bending can damage the coating and, therefore, WS_2_ is not suitable for large-angle bending in clinical applications [42,46]. In addition, it has demonstrated good biocompatibility in preliminary studies, but its cytotoxicity is still unproven, and there are no experiments simulating the intraoral environment, so subsequent studies need to focus on these aspects.

### 2.2. Zinc and Zinc Compound Coatings

ZnO is of interest because of its low toxicity and good biocompatibility. A human safety review showed that nanostructured ZnO is safe for humans [47], and its suitability for biomedical applications has led to an increasing number of studies on ZnO nanocoatings for reducing friction on fixed appliances. Mojghan Kachoei et al. conducted an in vitro study and found that coating SS archwires with ZnO nanoparticles resulted in a 39–51% [23] (using deposition-precipitation) or 64% [24] (using the sol-gel technique) reduction in friction force on the coated archwires compared to uncoated archwires, and this positive effect could be retained under variations in the angle between the slot and the archwire (0–10°) [23]. Another experiment showed an even better result: when the angle increased, its ability to reduce friction force was enhanced [25]. However, a similar study conducted by Ahmad Behroozian et al. [26] showed that there was no significant reduction in friction force after coating ZnO nanoparticles on SS archwires, but coating ceramic brackets with ZnO nanoparticles could reduce friction force. More interestingly, having both SS archwires and ceramic brackets coated with ZnO nanoparticles did not contribute to the reduction in friction force [26]. This phenomenon hints that when using SS archwires and ceramic brackets as a pair, there is no need to apply ZnO coatings on both the archwires and the brackets, as this would increase the cost and decrease the friction-reducing effect. In addition to SS, nickel titanium (NiTi) archwires also need reduced friction due to their wide usage. Coating ZnO nanoparticles on NiTi archwires using the chemical deposition method reduces surface friction by nearly 21% [25], while using electrochemical deposition reduces friction force by 34% [27].

ZnO NPs not only provide good biocompatibility and antibacterial properties, but also have better friction reduction than compact ZnO [23], so they should be prepared in the form of nanoparticles when the ZnO coating is performed. In addition, ZnO coatings prepared using the sol-gel technique can be applied chairside [24], making this one of the most feasible preparation methods in the clinic. Currently, many studies have demonstrated its ability to exert good friction reduction on 0.019 × 0.025 SS archwires [23,26]. In fact, it has worked well in various sizes and materials of archwires and brackets, including ceramic brackets, while its utility in self-ligating brackets has not yet been determined. However, pairing ZnO-coated SS archwires with ZnO-coated ceramic brackets is not recommended, as it is not effective in reducing friction [26].

The use of physical vapor deposition (PVD) for zinc (Zn) plating on the surface of SS archwires is thought to significantly reduce surface friction. Whether the angle between the archwire and the slot is 0° or 10°, the change in angle does not significantly affect this result [23]. In addition, tensile strength and three-point bending strength also increase significantly after Zn plating, indicating that Zn plating has a good effect on changing the mechanical properties of SS archwires and is one of the potentially available materials [48]. One of the advantages of Zn coating over ZnO coating is that fewer cracks are formed, and the mechanical properties are better maintained after bending, which indicates that it can be used in clinical situations where the archwire is to be bent at large angles.

### 2.3. Zirconia Compound Coatings

In clinical medicine, zirconium oxide (ZrO_2_) nanoparticle coatings are used on a variety of implant surfaces, such as artificial joints, skin implants, interventional catheters, and other surgical instruments, due to their high stability, corrosion resistance, and excellent biocompatibility [49,50,51]. Therefore, their use in orthodontic appliances has been investigated for performance improvements. However, among the NiTi, SS, and beta-titanium (TMA) archwires, ZrO_2_ could only be applied to the TMA wire, and no coating was observed on the other two archwires using scanning electron microscopy (SEM) [52]. Moreover, ZrO_2_ on the surface of TMA archwires did not have a significant effect on friction reduction, although it increased the surface smoothness of the archwires [52]. This differs from the outcomes of orthopedic implants studied in another field, which applied zirconia nanomaterials to titanium alloys and obtained lower friction [53]. This alloy has a different composition from TMA archwires, which could be the reason for the different results. Therefore, the composition of the substrate also needs to be taken into consideration when selecting the coating material. Based on the above conclusions, the clinical application of ZrO_2_ coatings in orthodontic appliances is of low value.

### 2.4. Titanium and Titanium Compound Coatings

The use of titanium compound coatings in orthodontics is becoming more widespread. Ion-plated TiN coatings have the characteristics of high hardness, wear resistance, corrosion resistance, and surface lubrication. They can form a passivated titanium dioxide (TiO_2_) layer on their surface, so TiN coatings are commonly used on various dental instruments and materials [54,55,56]. In an earlier study, the friction force of TiN-coated SS brackets coated via the hollow cathode discharge method and of uncoated SS archwires before and after corrosion in fluorinated solutions was investigated. It was found that the friction force of the coated brackets was not significantly lower than that of the uncoated brackets, whether before or after corrosion [28]. It has also been shown that TiN coated via ion beam-assisted deposition (IBAD) on SS materials can even increase the friction force [29]. Interestingly, a recent study by Arici N. et al. concluded that coating TiN using radio frequency magnetron sputtering on SS brackets can reduce the coefficient of friction (CoF) by 50% when applied in combination with uncoated SS archwires, while the combination of TiN-coated archwires and uncoated brackets increases the CoF whether the material of the archwires is NiTi or SS [30]. That is, when TiN is coated on SS brackets and archwires, there can be different influences on friction force. The different conclusions might be related to different coating methods or other conditions. Thus, more research is needed in order to determine and verify the reasons.

The effectiveness of TiN in reducing friction on the surface of SS archwires and brackets is still controversial. However, based on some of the studies where it was considered effective, it is easy to see that its clinical application requires pairing with specific brackets or archwires. For example, uncoated brackets are not recommended when TiN-coated archwires are used [30]. This suggests that attention should be paid to the application of TiN coatings when selecting brackets and archwires.

The current trend tends to combine TiN with other application materials. TiN-derived coating materials such as titanium carbonitride (TiCN) [31,32], multilayer titanium nitride/titanium (TiN/Ti) [33], and nickel-titanium-molybdenum (Ni-Ti-Mo) composite coatings [57] not only have a more exact friction reduction effect than TiN, but are also more beneficial to clinical application due to other advantages.

In a recent study, Suciu V et al. used direct-current (DC) reactive magnetron sputtering to cover the surface of SS archwires with TiN and TiCN films and varied the ambient nitrogen content during the film fabrication process to investigate the effects of different nitrogen contents on the properties of the produced films. They found that the higher the nitrogen and carbon contents, the lower the CoF. In particular, the TiCN coating could reduce the CoF to 0.30 and make the CoF more stable, while having better hardness and corrosion resistance than TiN. Therefore, overall, the TiCN film had better surface properties than TiN in the tested samples. More interestingly, the researchers also found that the colors of TiN and TiCN films have decorative aesthetic properties, so these two coatings may be among the future directions for the development of aesthetic friction reduction archwires [31]. Similarly, Jie Zhang et al. found that TiN and TiCN coatings on SS surfaces can reduce surface friction, especially in artificial saliva [32]. Liyuan Sheng et al. prepared different layers of TiN/Ti multilayer coatings on titanium-aluminum-vanadium alloy with 6% aluminum and 4% vanadium (Ti_6_Al_4_V) substrates with the same deposition time via high-power direct-current reactive magnetron sputtering, and they found that the CoF of 1- and 2-layer TiN/Ti multilayer coatings maintained an increasing trend, but the CoF of 4-, 8-, and 12-layer TiN/Ti multilayer coatings showed a decrease [33]. Thus, TiCN is superior to TiN in terms of friction reduction, hardness, corrosion resistance, and aesthetic properties, offering better clinical prospects. For multilayer coatings, researchers also found that in order to obtain the best clinical results, it is necessary to restrict the TiN/Ti coatings to no more than two layers. 

The above studies are all about SS archwires and brackets, while there are relatively few studies about titanium compound coatings on orthodontic archwires made from other materials. A study by Haruki Sugisawa et al. showed that TiN coatings can reduce the friction of NiTi and SS archwires, increase the tensile strength and stiffness of SS archwires, and reduce the elasticity of NiTi archwires [58]. In addition, Vinod Krishnan et al. attempted to apply a titanium aluminum nitride (TiAlN) coating to β-titanium orthodontic archwires using PVD to investigate its effect on friction, but the results showed no significant reduction compared to uncoated archwires [59]. Not many further studies on the usage of TiAlN coatings in orthodontic appliances have been followed up. In addition, a recent study investigated TiO_2_ and silicon dioxide (SiO_2_) nanoparticles on the surface of chromium-nickel (Cr-Ni) archwires and found that TiO_2_ did not show a significant ability to reduce friction [60]. Conversely, SiO_2_ can be used to reduce friction between the brackets and archwires under both dry and wet conditions, while making the surface of the archwires smoother than uncoated ones [60]. A systematic review and meta-analysis suggested that nanoparticle-coated orthodontic archwires can be considered to significantly reduce frictional resistance [61]. In this regard, it can be presumed that TiO_2_ nanoparticles are also theoretically able to reduce the friction force. In fact, the reason for the inability of TiO_2_ nanocoatings to significantly reduce friction may be that the coating is thicker than other coatings [60]. Therefore, the pressure on the inner wall of the slot is greater, resulting in greater friction force. In addition, the deformation and peeling of the surface morphology are also among the reasons [62]. Similarly, the application of TiO_2_ coatings on SS brackets using PVD does not reduce friction [63].

Therefore, the TiO_2_ coating may not exhibit the desired ability to reduce friction, because of its thickness [60]. However, is still worth developing as a very versatile material and, therefore, has a certain research value. In future research, other coating methods to reduce the thickness could be considered. Based on the consideration of film thickness, the preparation of a thinner film of TiO_2_ nanocoatings could be attempted in future studies. Recent studies have proven that low-pressure metal-organic chemical vapor deposition (LPMOCVD) can control the film thickness by adjusting the deposition time [64]. In fact, this method of controlling the thickness of the coating can be applied to many deposition-based coating methods; thus, future research could take it into consideration when finding the most suitable coating conditions.

In addition to titanium compounds, composite coatings of titanium with other metals are also being investigated. In the study of Yılmaz H et al. (2021), nickel ions and TiO_2_ nanoparticles were used as standard coating materials on the surface of NiTi archwires, while the addition of 0.0015 g/dm^3^ sodium molybdate (Na_2_MoO_4_) on this basis led to the formation of a Ni-Ti-Mo coating on the surface of the archwires; thus, a reduction in the CoF from 0.288 to 0.252 was observed. However, although the addition of a certain amount of molybdenum ion (Mo^6+^) to the bath solution can reduce the surface friction of the archwires after the formation of the coating, the reduction in surface friction gradually becomes less effective as the concentration of Mo^6+^ increases. The standard NiTi coating does not reduce friction, and the addition of different concentrations of chromium (Cr) to the standard coating material to form a nickel-titanium-chromium (Ni-Ti-Cr) coating also does not reduce friction. [57]

In summary, the trends of Ti compound coatings are nitriding [30], incorporation of other metal ions [57], multi-layered materials [33], and low-thickness materials [64]. Most of these are still some distance away from being applied in clinical practice, but attempts can be made to find the most suitable preparation conditions and methods to obtain coating materials with satisfactory properties in all respects. The biggest disadvantage of this type of coating, however, may be its high price [65]. If the price of orthodontic supplies is increased by the application of the coating, then there is no advantage compared to other coatings with similar properties, and patients are less likely to choose to use these materials. Therefore, although the materials are excellent in various respects, in order to put them into use, there is still a need to find ways to reduce the cost of their production.

### 2.5. Silver and Silver Compound Coatings

Nanosilver coatings can reduce the surface roughness of SS brackets [63] or NiTi archwires [66], but the change in friction between brackets and archwires is not obvious [63]. Moreover, silver (Ag) plating on 0.019 × 0.025-inch coated SS archwires significantly reduces the friction between the brackets and the archwires, while 0.017 × 0.025-inch SS archwires do not significantly reduce the friction [67]. The 0.019 × 0.025-inch SS archwires, whether coated or not, have higher friction than the 0.017 × 0.025-inch SS archwires. The play between 0.019 × 0.025-inch SS archwires and the slots is smaller, so the pressure is higher when the slots are of the same size. After applying the coating, the friction coefficient of the surface should be the same regardless of the size of the archwires, so the alteration of the friction force of the archwires subjected to more pressure may be relatively more obvious when the other conditions are the same [67]. However, this explanation has not yet been clearly confirmed. Therefore, more research is still needed in order to confirm its capabilities. Still, considering the antibacterial properties of silver [63], nanosilver coatings are still a popular research topic for surface coatings of orthodontic appliances, even without clear friction reduction properties.

Currently, there are only a few types of materials related to silver used in the surface coating of orthodontic appliances, which are generally silver nanoparticles (AgNPs). Interestingly, Jia Wang et al. [68] successfully prepared silver-doped tantalum boride (TaAgB) solid solution coatings on SS blocks using magnetron sputtering technology with Ag doping. They verified the effectiveness of the coating in reducing the CoF of the SS material to 0.08, as shown in Figure 3, and suggested that it could be used for orthodontic archwires and brackets because of its good hardness, strength, and biocompatibility. Therefore, as a surface coating material with low friction, good antibacterial properties, high wear resistance, and high corrosion resistance, it has strong potential value for application on fixed appliances.

As a precious metal, silver coatings are also a relatively high-value material. However, nanosilver coatings have multifunctional properties that may be accepted by the general public even if their price is high [69]. Therefore, it is necessary to develop more friction reduction, antibacterial, and other properties of nanosilver coatings.

### 2.6. Aluminum and Aluminum Compound Coatings

Al_2_O_3_ coatings are oxide coatings that are common in various fields. Research by Nursel Arici et al. [30] in 2021 demonstrated the friction reduction effect of Al_2_O_3_ coatings in detail. When an Al_2_O_3_ coating is applied on the surface of SS brackets, whether the material of the combined archwire is made of SS or NiTi, and regardless of whether the archwires have an Al_2_O_3_ coating or not, the CoF can be reduced significantly. Among all combinations, the combination of Al_2_O_3_-coated SS brackets with the same coated archwire was able to obtain the lowest CoF. That is, the combination of coated SS brackets-coated SS archwire is better than other combinations. It is also worth mentioning that in almost all cases involving coated archwire-uncoated bracket combinations, an increase in CoF was observed for both NiTi and SS archwires, being much higher than for both uncoated combinations. This phenomenon may be related to the peeling of the coating [30]. Therefore, when testing friction, the pressure of the coating environment, time, immersion conditions, and peeling should also be taken into account to obtain more accurate conclusions.

Others have combined aluminum (Al) and SiO_2_ to form Al-SiO_2_ composite coatings. The excellent friction reduction ability of SiO_2_ has been mentioned above [60]. In contrast, the friction reduction properties of Al have not been adequately studied. A reduction in the CoF is observed in Al-SiO_2_-coated NiTi and SS archwires compared to the uncoated ones [70]. Therefore, Al-SiO_2_ composite coatings may also be used to reduce the friction of orthodontic archwires in the future.

Al compound coatings are effective in reducing friction, retain high aesthetic value, and are relatively inexpensive [30]. However, studies about them are still scarce, especially since their performance changes after long-term use are not confirmed. In addition, Al_2_O_3_ needs to be considered in clinical applications for the combination of coated and uncoated archwires and brackets. Combining coated archwires with uncoated brackets is not recommended, but it is possible to use coated brackets only [30]. Ideally, both the brackets and the archwire should be coated, which will result in the best friction reduction.

### 2.7. Other Metal and Metallic Compound Coatings

Nursel Arici et al. [30] noted that the CrN coating was ineffective in reducing friction between the SS brackets and the SS or NiTi archwires, whether both were coated or only one was coated. SEM images after friction tests showed that the CrN coating peels off from the surface. Therefore, residual coating material at the contact area between the surfaces may be responsible for the increase in CoF. This phenomenon is not found in the application of hard chrome carbide plating (HCCP) coatings. HCCP coatings significantly reduce the friction of SS archwires, and mechanical properties such as flexural strength and flexural modulus remain unchanged [21]. HCCP is a promising coating material that can potentially be used clinically to reduce friction. However, its shortcomings include aesthetics that need to be improved and biocompatibility that has not been explicitly tested.

Another common metal coating is rhodium. As a coating based on aesthetic considerations, it is generally applied to archwires. Some recent studies have begun to explore its usage beyond aesthetics. A study by Tahereh Hosseinzadeh Nik et al. found that the frictional force generated during sliding between ceramic brackets and rhodium-plated SS archwires was significantly less than that of ceramic brackets and SS archwires. Even after immersion in 0.05% sodium fluoride (NaF) mouthwash, the increase in friction was much less in the former than in the latter [71]. This indicates that rhodium plating can be used not only as a method to increase the aesthetic effect of the archwires, but also to reduce the friction between the archwires and the brackets, constituting a method to increase the efficiency of the orthodontic treatment and maintain good performance even when the patient uses NaF mouthwash. In addition, they demonstrated that rhodium-plated SS archwires showed a smaller increase in roughness than uncoated SS archwires after treatment in mouthwash [71]. Of course, the correlation between surface roughness and friction remains controversial. Some think that the two can be directly correlated [72,73], while others do not [35,74], so it is not yet possible to conclude that the smaller increase in friction of rhodium-plated archwires is related to the smaller increase in surface roughness. Friction-reducing coatings made of other metallic monomers are also gradually being investigated. Recent studies have attempted to prepare niobium (Nb), tantalum (Ta), and vanadium (V) coatings on 316 SS substrates using the plasma sputtering method and found that V coatings have an effect on the roughness of the SS surface [75]. However, other experiments on the tribology of these new materials have not yet been conducted.

### 2.8. Carbon-Based Coatings

Carbon-based coating materials mainly contain diamond-like carbon (DLC) and graphene.

DLC is a diamond-like carbon film that exhibits extremely high hardness, a low coefficient of friction, chemical inertness, and high corrosion resistance [34]. Research has demonstrated the biomedical applications of DLC films. For example, implants with protective films can extend the life of the implant by reducing corrosion and wear, which can benefit patients [76].

DLC-coated SS brackets exhibit lower static [77] and dynamic friction [78], and their effect is better than that of TiN-coated brackets [79]. However, some studies have also found no reduction in friction on the surface of DLC-coated SS brackets [80]. In addition to brackets, DLC-coated SS archwires [78,80,81,82] or NiTi archwires [83,84] can be applied to reduce friction. Hao Zhang et al. concluded that the use of the plasma-enhanced chemical vapor deposition (PECVD) method to cover the surface of SS archwires with a DLC coating can reduce the coefficient of dynamic friction by 40.71% and significantly improve the surface hardness of the archwire, as well as reducing friction and exhibiting good biocompatibility [82]. It has also been shown that the reduction in friction is effective whether the angle between the archwire and the bracket is 0° or 10° [81].

Furthermore, the application of DLC via different coating methods—such as plasma-enhanced chemical vapor deposition (PECVD) [79,82,85], plasma-based ion implantation/deposition (PBIID) [81,84,86], and mirror confinement electron cyclotron resonance (MCECR) plasma sputtering [80,83]—has shown similar friction-reducing effects. It has also been shown that for DLC-coated SS brackets, the size and material of the archwire, the contact angle, and the dryness or humidity of the experimental conditions have no influence on the friction reduction effect of DLC coatings [77]. However, although changes in the above conditions have no significant influence on the effect of DLC coatings in reducing friction, the effect of the thickness of the coating is still not negligible. Muguruma et al. [78] investigated the mechanical properties of DLC coatings of different thicknesses. They found that a thin DLC coating reduces surface friction, which is the same result as previously described, yet the relatively thick DLC layer tends to break from the DLC-steel interface. In addition, thicker layers have a negative effect on friction force. DLC coatings also have the advantage of being resistant to high-fluorine environments. Even at high fluoride ion concentrations, the friction force remains relatively stable, i.e., there is no significant increase [83].

DLC coatings with different properties were prepared under different conditions as DLC-1 and DLC-2. DLC-1 uses a 10 kV target voltage, an acetylene-toluene gas environment, and deposits for 3 min to form a structure with a diamond-rich outer surface and a graphite-rich inner surface; DLC-2 uses a 7 kV target voltage, a toluene gas environment, and deposits for 4 min to form a structure with a graphite-rich outer surface and a diamond-rich inner surface. The results showed that DLC-2 produces significantly lower friction than the uncoated case when wet and at an angle of 10°, which was not observed for DLC-1. The reasons for this were that partial cracks in the DLC-1 coating led to increased friction, while no cracks or breakages of the coating were observed in DLC-2. Therefore, we can conclude that the DLC-2 coating is more valuable than DLC-1 in practical applications because of its good flexibility and adhesion [87].

In addition to pure DLC, fluorine-doped DLC (F-DLC) and silicon-doped DLC (Si-DLC) coatings can also reduce friction on the surface of SS brackets. The F-DLC-coated brackets exhibit low static friction between the brackets and the archwire under moist conditions, which is lower than that under dry conditions. However, doping DLC with fluorine or silicon results in a significant reduction in surface hardness. Even though F-DLC is most effective in reducing the friction between the archwire and the brackets in humid conditions, this disadvantage needs to be taken into account in practical applications [85]. Titanium-doped DLC (Ti-DLC) coatings are also ideal friction-reducing films for orthodontic brackets, as they exhibit a lower CoF than TiCN and TiN in most cases, under both artificial saliva and dry conditions [32].

Recently, research on graphene coatings has gradually emerged. Commonly used graphene materials include graphene oxide and graphene-sheet-embedded carbon (GSEC). Graphene oxide (GO) coatings can reduce the CoF of NiTi alloy [88,89], while the addition of AgNPs and the formation of graphene oxide/silver nanoparticle (GO/Ag) coatings can further reduce CoF [88]. In artificial saliva, the GSEC surface coating of SS archwires coated with MCECR reduced the CoF to 0.12 [90]. Pengfei Wang et al. later discovered ways to make the friction on the surface of GSEC coatings even lower. In this study, they found that the thickness of the coating increased as the GSEC film deposition time increased. After 80 min of deposition, both stable and low friction coefficients could be obtained, even reaching a level of less than 0.10 when running against three-row microgroove-textured SS brackets in an artificial saliva environment [46]. The possible mechanisms are shown in Figure 4. Although immersion causes a rebound in CoF, after 30 days of immersion, the CoF remains below 0.30, consistently exhibiting very good friction reduction [90]. This illustrates the effectiveness of the graphene coating in reducing the friction on the surface of the archwire.

In summary, carbon-based friction-reducing coatings have good overall durability and biocompatibility [82], and DLC coatings are resistant to corrosion by fluoride-containing mouthwashes [83]. The coating method and conditions are less restrictive compared to metal coatings, but there are strict requirements for the thickness of the coating layers, requiring a thickness that is both thin and effective [87]. GSEC coatings can be used for up to one month, and the optimal coating deposition length has been determined and may be ready for clinical application soon.

### 2.9. Polymeric and Bioactive Coatings

Teflon (polytetrafluoroethylene (PTFE)), an aesthetic white coating for dental instruments, has been shown to provide a smoother surface and reduce frictional resistance and frictional damage [35,36,37,38,39] on NiTi [91] or SS archwires [92]—especially when paired with metal slots in ceramic brackets [38]—in both dry and wet conditions. However, when combined with ceramic brackets, PTFE- and epoxy-coated NiTi archwires will increase the friction between the archwires and the brackets [93]. This is probably because ceramics have a rougher texture and a more porous surface than stainless steel. Another research shows the contrast, which whether combined with ceramic or metal brackets, PTFE-, or epoxy-coated SS archwires always show higher friction resistance [94]. However, the latest study by K. Ranjan and R. Bhat et al. contradicts this finding; the study concludes that whether paired with SS brackets, ceramic brackets, or ceramic brackets with metal slots, the PTFE-coated archwire produces lower frictional resistance compared to uncoated SS archwires [92]. In addition, research has investigated suitable preparation conditions. PTFE coatings prepared at 200 °C offer lower friction, higher wear resistance, lower coloration, and higher resistance to microbial adhesion than those prepared at 380 °C [95].

In addition to PTFE, other polymer surface coatings have been used to explore the ability to reduce friction on fixed appliances, such as chitosan (CTS) [24], 2-methacryloyloxyethyl phosphorylcholine (MPC), parylene, and epoxy.

Elhelbawy N. et al. investigated CTS as a newer and potentially useful coating material for reducing friction on fixed appliance surfaces in comparison to ZnO. This comparison revealed no significant difference in the degree of friction reduction between CTS and ZnO (64 and 53%, respectively). Interestingly, when CTS and ZnO were applied to the brackets or archwires alone, and the matching archwires or brackets were not coated, the degree of friction reduction was similar to that of having both brackets and archwires coated with CTS or ZnO, suggesting that coating both the brackets and the archwires did not significantly improve this aspect of friction reduction [24].

In a recent study, Ryo Kunimatsu et al. found a promising application of MPC in fixed appliances. They covered SS archwires with an MPC coating and performed tensile tests using a bracket-archwire combination to demonstrate the coating’s ability to reduce friction. Furthermore, they directly measured the efficiency of tooth movement using an in vitro experimental tooth movement model at 50 g and 100 g traction force, and finally confirmed that the SS archwire coating with MPC improves the efficiency of tooth movement by reducing the occurrence of friction [96]. This experimental design has provided ideas for subsequent studies. Simply confirming that a certain surface coating can reduce friction is not enough to fully prove its clinical value. Its positive effect on improving tooth movement efficiency needs to be further determined through model experiments, animal experiments, and even clinical trials.

Chin-Yu Lin et al. [97] coated SS archwires with parylene, epoxy, or PTFE and tested the friction between the archwires and ceramic brackets after immersion in water for different lengths of time. Compared to the other groups, the epoxy-coated archwires presented higher or no significant difference in friction resistance and were the only coated archwires whose maximal resistance to sliding at a 3° contact angle (MRS3) was consistently lower than that of the uncoated control group. Parylene, on the other hand, had relatively high friction at all water immersion times (0 to 4 weeks) and angles (0 to 3°). Therefore, epoxy-coated archwires may even be a better choice than PTFE.

In addition to polymers, bioactive glass (BG) coatings have recently been prepared on the surface of SS archwires using electrophoretic deposition. Although BG has higher hardness, elastic modulus, and surface smoothness, it failed to reduce the friction in the experiment [40]. Overall, studies on the application of bioactive coatings on fixed appliances are extremely rare and could perhaps be explored more in the future.

Carbon-based materials [82], natural polymeric materials [24], and bioactive materials [40] are safer than metal and metallic compound materials and can be used to replace other coating materials that are potentially cytotoxic [42]. CTS coatings can replace ZnO and are relatively more convenient and cost-effective, as they do not require coating on both the archwire and the bracket surface [24]. Aesthetic coatings such as PTFE are suitable for both SS and NiTi archwires, but when used with ceramic brackets, it is better to choose ceramic brackets with metal slots [92] and other commonly used polymeric aesthetic archwires. Among the other common aesthetic polymer coatings on the market, only epoxy shows a stronger friction reduction effect than PTFE [97]. However, the popularity of its use and the difficulty of its preparation have not been conclusively proven, so the most suitable polymer aesthetic coating for clinical friction reduction is still PTFE.

Recent studies on friction reduction coatings for orthodontic appliances are summarized in Table 1.

**Table 1 ijms-24-06919-t001:** Recent studies on friction reduction coatings for orthodontic appliances.

Ref.	Coating Materials	Coating Technique	Substrate	Study Type	Roughness	Friction Reduction Effectiveness	Wear Mechanism	Wear Resistance	Other Effectiveness
[67]	Ag	Direct current sputtering	SS archwires	In vitro (dry condition)	NA	Compared to uncoated archwires, the friction force of 0.019 × 0.025-inch SS archwires reduced 1 N after coating (*p* = 0.032), but no significant difference between coated and uncoated 0.017 × 0.025-inch SS archwires (*p* = 0.854).	NA	NA	NA
[30]	Al_2_O_3_	Radio frequency magnetron sputtering	SS brackets, NiTi archwires, SS archwires	In vitro (dry condition)	Surface roughness decreased with the coating process. R_a_ = 331.53 nm, R_q_ = 426.17 nm (coated brackets); R_a_ = 361.64 nm, R_q_ = 466.01 nm (coated NiTi archwires); R_a_ = 95.86 nm, R_q_ = 128.01 nm (coated SS archwires)	NiTi archwires: When both brackets and archwires were coated and only brackets were coated, CoF reduced from 0.316 to 0.238 and 0.251, respectively. CoF increased from 0.316 to 0.400 when only archwires were coated. SS archwires: When both brackets and archwires were coated, only brackets were coated, and only archwires were coated, CoF reduced from 0.552 to 0.227, 0.2,35, and 0.445, respectively.	No peeling off	The coatings did not peel off after friction, thermal, and brushing tests.	NA
[70]	Al-SiO_2_	Magnetron sputtering	NiTi and SS archwires	In vitro (in artificial saliva)	NA	CoF of NiTi archwires and SS archwires reduced from 0.68 to 0.46 and from 0.58 to 0.45, respectively.	NA	Corrosion-resistant	Corrosion-resistant effectiveness Biocompatibility
[40]	BG	Electrophoretic deposition	SS archwires	In vitro (dry condition)	Sa increased from 0 to 0.46–0.79.	Friction forces showed no significant reduction.	NA	None of the coatings were damaged after three-point bending and they sustained good interfacial adhesion.	Aesthetic effect
[84]	CF_4_	Plasma-based ion implantation and deposition	NiTi archwires	In vitro (dry condition)	NA	Friction forces reduced from 129.48 to 104.97 gf, which showed no significant reduction.	NA	NA	NA
[84]	CH_4_	Plasma-based ion implantation and deposition	NiTi archwires	In vitro (dry condition)	NA	Friction forces reduced from 129.48 to 87.30 gf, which showed no significant reduction.	NA	NA	NA
[29]	CN_x_	Ion beam-assisted deposition	304L SS disks	In vitro (friction test: NA; antibacterial test: in bacterial suspension)	The films slightly reduce the surface roughness parameter. R_a_ reduced from 0.181 to 0.140 μm	CoF reduced from 0.431 to 0.188 (*p* < 0.05).	NA	NA	Antibacterial Effectiveness Biocompatibility
[30]	CrN	Radio frequency magnetron sputtering	SS brackets, NiTi archwires, SS archwires	In vitro (dry condition)	Surface roughness decreased with the coating process. R_a_ = 276.85 nm, R_q_ = 360.24 nm (coated brackets); R_a_ = 354.35 nm, R_q_ = 454.66 nm (coated NiTi archwires); R_a_ = 190.28 nm, R_q_ = 229.26 nm (coated SS archwires)	NiTi archwires: When both brackets and archwires were coated, only brackets were coated, and only archwires were coated, CoF increased from 0.316 to 0.443, 0.324, and 0.505, respectively. SS archwires: When both brackets and archwires were coated and only archwires were coated, CoF increased from 0.552 to 0.598 and 0.586, respectively. CoF reduced from 0.552 to 0.410 when only brackets were coated.	Peeling off	Large areas of peeling could be seen after friction, thermal, and brushing tests.	NA
[24]	CTS (NPs)	Sol-gel dip coating	SS archwires and SS brackets	In vitro	NA	Friction force decreased by ~53%.	NA	NA	NA
[84]	DLC	Plasma-based ion implantation and deposition	NiTi archwires	In vitro (dry condition)	NA	The friction force reduced from 129.48 to 86.13 gf (*p* = 0.039).	NA	NA	NA
[87]	DLC	Plasma-based ion implantation and deposition	SS archwires	In vitro (in artificial saliva and dry condition)	NA	When coated with DLC-2, the static friction reduced from 2.39 to 2.09 N in artificial saliva and from 2.49 to 2.25 N under dry conditions, and the kinetic friction reduced from 2.37 to 1.99 N in artificial saliva and from 2.55 to 2.21 N under dry conditions, which showed a significantly lower frictional force than the uncoated archwires, while DCL-1 showed no significant difference compared with uncoated samples.	Rupture	No rupture was observed for the DLC-2 condition after the drawing-friction testing.	NA
[97]	Epoxy	NA (commercial)	SS archwires	In vitro (in distilled-deionized water)	NA	The average resistance under 0° bracket-wire angle reduced from 1.63 to 1.13 N immediately after being coated. The average resistance under a 3° bracket-wire angle reduced from 5.12 to 4.27 N immediately after being coated.	NA	NA	Durability (>4 weeks)
[94]	Epoxy	NA (commercial)	SS archwires	In vitro (in artificial saliva and dry condition)	NA	The friction force increased from 3.00–9.00 N to 16.00–20.50 N in both wet and dry conditions.	NA	NA	NA
[93]	Epoxy	NA (commercia)	NiTi archwires	In vitro (dry condition)	NA	Friction forces increased from 49.287 to 53.316 gf.	NA	NA	NA
[89]	GO	Silane coupling	NiTi archwires	In vitro (friction test: NA; antibacterial test: in bacterial suspension; corrosion test: artificial saliva)	The surface of the samples coated with 2 mg/mL GO concentrations was smooth with a uniformly coated area.	CoF reduced from ~0.9 to 0.2–0.4.	Grooves in the same direction as the gliding	The samples coated with 0.5 mg/mL GO concentrations had fewer grooves, but a small amount of wear debris was present.	Antibacterial effectiveness Corrosion-resistant effectiveness Biocompatibility
[88]	GO/Ag (NPs)	Electrophoretic deposition	NiTi alloy	In vitro (dry condition)	R_a_ ranged from 50.72–69.93 nm.	CoF reduced from 0.060 to 0.006, and increased coating time led to lower CoF.	NA	NA	NA
[46,90]	GSEC	Electron cyclotron resonance plasma sputtering	SS archwires	In vitro (in artificial saliva)	NA	CoF reduced to ~0.10 and remained under 0.30 after 30 days.	Peeling off	The corresponding wear rate was strongly decreased from 4.84 × 10^−6^ to 0.11 × 10^−6^ mm^3^/Nm.	NA
[21]	HCCP	Electroplating	SS archwires	In vitro (in PBS solution and dry condition)	There were very small protrusions on the surfaces of the coated archwires, while the surfaces of uncoated archwires were smooth.	The friction force reduced from 147.15 to 124.61 gf (*p* = 0.0076) and from 143.55 to 121.41 gf (*p* = 0.04) under dry and wet conditions, respectively.	Scratches	After the friction test, scratches were seen on the surfaces on the coated surfaces.	Aesthetic effect
[18]	Ni+MoS_2_ (NPs)	Electrochemical co-deposition	SS archwires	In vitro (in artificial saliva and dry condition)	NA	Dry conditions: CoF reduced from 0.58–1.43 to 0.50–1.19. In artificial saliva: CoF reduced from 0.95–2.52 to 0.94–2.35.	NA	NA	NA
[18]	Ni+WS_2_ (NPs)	Electrochemical co-deposition	SS archwires	In vitro (in artificial saliva and dry condition)	NA	Dry conditions: CoF reduced from 0.58–1.43 to 0.42–1.06. In artificial saliva: CoF reduced from 0.95–2.52 to 0.66–1.46.	NA	NA	NA
[57]	Ni-Ti-Cr	Chronopotentiometry	NiTi archwires	In vitro (in artificial saliva)	NA	CoF showed no significant reduction.	NA	NA	Corrosion-resistant effectiveness Durability
[57]	Ni-Ti-Mo	Chronopotentiometry	NiTi archwires	In vitro (in artificial saliva)	NA	CoF reduced from 0.288 to 0.252–0.265.	NA	NA	Corrosion-resistant effectiveness Durability (>60 days)
[97]	Parylene	NA (commercial)	SS archwires	In vitro (in distilled-deionized water)	NA	The average resistance under 0° bracket-wire angle increased from 1.63 to 5.39 N immediately after being coated. The average resistance under the 3° bracket-wire angle increased from 5.12 to 11.38 N immediately after being coated.	NA	NA	Durability
[97]	PTFE	NA (commercial)	SS archwires	In vitro (in distilled-deionized water)	NA	The average resistance under 0° bracket-wire angle reduced from 1.63 to 1.15 N immediately after being coated. The average resistance under a 3° bracket-wire angle showed no significant change immediately after being coated.	NA	NA	Durability
[95]	PTFE	Spraying	SS, Ni-Ti, and β-titanium archwires	In vitro	R_a_ of coated archwires increased from 0.02–0.21 to 0.53–0.58 μm.	The friction force of coated archwires reduced from 123.94–152.61 to 102.98–124.40 gf compared to uncoated archwires and 200 °C coating resulted in less friction against brackets than did the conventional 380 °C coating.	Scratches	PTFE-coating at 200 °C resulted in good microbial adhesion and tolerance of wear.	Durability (>3 months) Aesthetic effect
[94]	PTFE	NA (commercial)	SS archwires	In vitro (in artificial saliva and dry condition)	NA	The friction force was higher than uncoated archwires but lower than epoxy and rhodium-coated archwires in wet and dry conditions. Dry conditions: The friction force increased from 3.80–9.00 to 5.50–12.80 N. In artificial saliva: The friction force increased from 3.00–8.20 to 3.80–11.60 N.	NA	NA	NA
[93]	PTFE	NA (commercial)	NiTi archwires	In vitro (dry condition)	NA	Friction forces increased from 49.287 to 61.427 gf.	NA	NA	NA
[92]	PTFE	NA (commercial)	SS archwires	In vitro	The surface roughness increased after the friction test.	When compared with ceramic with a metal slot, the friction force reduced from 1.61 to 1.03 N after the archwires were coated with PTFE.	NA	NA	NA
[21]	Rhodium	NA (commercial)	SS archwires	In vitro (in PBS and dry condition)	There were protrusions on the surfaces of the coated archwires, while the surfaces of uncoated archwires were smooth.	The friction force increased from 147.15 to 216.29 gf (*p* < 0.001) and from 143.55 to 210.21 gf (*p* < 0.001) under dry and wet conditions, respectively.	Scratches	After the friction test, scratches were seen on the surfaces on the coated surfaces.	Aesthetic effect
[94]	Rhodium	NA (commercial)	SS archwires	In vitro (in artificial saliva and dry condition)	NA	The friction force increased from 3.00–9.00 N to 17.30–21.70 N in both wet and dry conditions.	NA	NA	NA
[71]	Rhodium	NA (commercial)	SS archwires	In vitro (in artificial saliva and 0.05% NaF mouthwash)	Surface roughness showed no significant reduction with the coating process.	The friction force was reduced from 2.22 to 1.49 N in artificial saliva and reduced from 2.72 to 2.17 N in 0.05% sodium fluoride mouthwash.	NA	NA	NA
[60]	SiO_2_ (NPs)	NA (commercial)	Cr-Ni archwires	In vitro (in artificial saliva and dry condition)	Surface roughness decreased with the coating process. R_a_ reduced from 0.673 to 0.040 μm.	Friction force reduced from 0.51 to 0.38 N and from 0.56 to 0.44 N under wet and dry conditions, respectively.	NA	NA	NA
[68]	TaAgB	Magnetron sputtering with Ag-doping	304 SS sheets	In vitro (in artificial saliva)	NA	CoF reduced from 0.38 to 0.08.	No serious wear phenomenon	The wear rate was 6.51 × 10^−15^ m^3^/Nm.	Antibacterial effectiveness Biocompatibility
[32]	TiCN	PVD	316L SS plates	In vitro (in artificial saliva and dry condition)	R_a_ = 29.86 nm; root mean square roughness: R_q_ = 39.09 nm	CoF reduced from 0.20 to less than 0.06 under dry conditions with loads of 5 N.	NA	NA	NA
[31]	TiCN	Direct current reactive magnetron sputtering	316L SS and (100)-oriented Si substrates	In vitro (in artificial saliva)	NA	CoF reduced to less than 0.30, and the addition of N_2_ and C in the Ti matrix further lowered the CoF.	NA	The lowest wear rate was 5.6 × 10^−6^ mm^3^/Nm	Corrosion-resistant effectiveness Aesthetic effect
[32]	Ti-DLC	PVD	316L SS plates	In vitro (in artificial saliva and dry condition)	The surface of the Ti-DLC film has many nanocrystal clusters that cause substantial pitting and layer defects. R_a_ = 10.40 nm; R_q_ = 13.43 nm	CoF remained less than 0.04 under both dry and artificial saliva conditions.	NA	NA	NA
[29]	TiN	Ion beam-assisted deposition	304L SS disks	In vitro (friction test: NA; antibacterial test: in bacterial suspension)	The films slightly reduce the surface roughness parameter. R_a_ reduced from 0.181 to 0.162 μm.	CoF increased from 0.431 to 0.469 (*p* < 0.05).	NA	NA	Antibacterial Effectiveness Biocompatibility
[58]	TiN	Ion plating	SS and NiTi archwires	In vitro (friction test: NA; corrosion test: in 0.9% NaCl solution)	R_a_ of TiN-coated SS archwire increased from 0.023 to 0.046 µm. R_a_ of TiN-coated NiTi archwire remained at 0.001 µm.	At angles of 0 degrees, the friction forces of the TiN-coated NiTi archwires reduced, but SS archwires showed no significant difference. At angles of 10 degrees, the friction forces of the TiN-coated SS and NiTi archwires were both reduced.	NA	NA	Corrosion-resistant effectiveness
[32]	TiN	PVD	316L SS plates	In vitro (in artificial saliva and dry condition)	The surface of the TiN film has some particles and pinholes. R_a_ = 26.59 nm; R_q_ = 32.62 nm	CoF reduced from 0.08 to less than 0.03 under dry conditions with loads of 5 N.	NA	NA	NA
[30]	TiN	Radio frequency magnetron sputtering	SS brackets, NiTi, and SS archwires	In vitro (dry condition)	R_a_ = 354.12 nm, R_q_ = 458.74 nm (coated brackets); R_a_ = 391.99 nm, R_q_ = 534.36 nm (coated NiTi archwires); R_a_ = 105.15 nm, R_q_ = 132.67 nm (coated SS archwires)	NiTi archwires: When both brackets and archwires were coated, only brackets were coated, and only archwires were coated, CoF increased from 0.316 to 0.399, 0.331, and 0.446, respectively. SS archwires: When both brackets and archwires were coated and only brackets were coated, CoF reduced from 0.552 to 0.372 and 0.237, respectively. CoF increased from 0.552 to 0.818 when only archwires were coated.	Peeling off	After friction, thermal, and brushing tests, the coatings peeled off in some small areas.	NA
[31]	TiN	Direct current reactive magnetron sputtering	316 L SS and (100)-oriented Si substrates	In vitro (in artificial saliva)	NA	CoF was reduced to less than 0.30, and the addition of N_2_ in the Ti matrix further lowered the CoF.	NA	The lowest wear rate was 7.8 × 10^−6^ mm^3^/Nm	Corrosion-resistant effectiveness
[29]	TiO_2_	Ion beam-assisted deposition	304L SS disks	In vitro (friction test: NA; antibacterial test: in bacterial suspension)	NA	NA	NA	The TiO_2_ film wore out in a few seconds.	Antibacterial effectiveness Biocompatibility
[60]	TiO_2_ (NPs)	NA (commercial)	Cr-Ni archwires	In vitro (in artificial saliva and dry condition)	Surface roughness decreased with the coating process. R_a_ reduced from 0.673 to 0.042 μm.	Friction force showed no significant reduction or even increased from 0.56 to 0.61 N compared to uncoated archwires.	NA	NA	NA
[48]	Zn	PVD	SS archwires	In vitro	NA	The friction force reduced from 2.98 to 2.03 N (*p* = 0.001) and from 3.51 to 1.72 N (*p* < 0.0001) at 0° and 10° angles, respectively.	Cracks and scratches	The scratches and cracks could be seen clearly.	NA
[46]	ZnO (NPs)	Sol-gel dip coating	SS archwires and SS brackets	In vitro	NA	Friction force decreased by ~64%.	NA	NA	NA
[27]	ZnO (NPs)	Electrochemical deposition	NiTi archwires	In vitro (friction test: NA; antibacterial test: on nutrient agar plates)	NA	Friction force decreased by 34%.	NA	NA	Antibacterial effectiveness
[75]	Niobium (NPs)	Plasma sputtering	316L SS	In vitro	R_a_ reduced from 157.8 to 133.3–156.1 nm after being coated (*p* > 0.05).	Potential friction reduction effectiveness	NA	NA	NA
[75]	Tantalum (NPs)	Plasma sputtering	316L SS	In vitro	R_a_ reduced from 157.8 to 110–130.6 nm after being coated (*p* > 0.05).	Potential friction reduction effectiveness	NA	NA	NA
[75]	Vanadium (NPs)	Plasma sputtering	316L SS	In vitro	R_a_ reduced from 157.8 to 83.4–96.8 nm after being coated (*p* = 0.002).	Potential friction reduction effectiveness	NA	NA	NA

Ag: silver; NA: not applicable; SS: stainless steel; NiTi: nickel titanium; R_a_: average roughness; R_q_: root mean square deviation; CoF: coefficient of friction; Al-SiO_2_: aluminum-silicon dioxide; BG: bioactive glass; S_a_: mean surface roughness; CF_4_: tetrafluoromethane; CN_x_: carbon nitride; NPs: nanoparticles; CTS: chitosan; DLC: diamond-like carbon; DLC-1: a DLC structure with a diamond-rich outer surface and a graphite-rich inner surface; DLC-2: a DLC structure with a graphite-rich outer surface and a diamond-rich inner surface; GO/Ag: graphene oxide/silver; GSEC: graphene-sheet-embedded carbon; HCCP: hard chrome carbide plating; PBS: phosphate-buffered solution; Ni: nickel; MoS_2_: molybdenum disulfide; WS_2_: tungsten disulfide; Ni-Ti-Mo: nickel-titanium-molybdenum; Ni-Ti-Cr: nickel-titanium-chromium; PTFE: polytetrafluoroethylene; SEM: scanning electron microscopy; SiO_2_: silicon dioxide; Cr-Ni: chromium-nickel; TaAgB: silver-doped tantalum boride; TiCN: titanium carbonitride; PVD: physical vapored deposition; Si: silicon; N_2_: nitrogen; C: carbon; Ti: titanium; Ti-DLC: titanium-doped DLC; TiN: titanium nitride; TiO_2_: titanium dioxide; ZnO: zinc oxide; Zn: zinc; NaF: sodium fluoride.

## 3. Antibacterial Coatings

Among orthodontic antimicrobial coatings, the most valued are Ag and ZnO coatings, as they have been shown to exert good antimicrobial activity and are safe for humans in other areas of medicine [98,99]. Recently, TiO_2_ photosensitive antimicrobial coatings, polymeric coatings originally used for aesthetic purposes, and bioactive lysozyme coatings have also gradually entered the trend and have been explored for their potential as antibacterial coatings. As for the research on the antimicrobial properties of carbon-based material coatings, only Dai et al. have investigated this topic, and the material used was GO [89]. This study found that when the GO concentration was low, the GO coating could not completely cover the NiTi substrate, and the tribological and anticorrosion properties were barely improved, while the antibacterial properties—although statistically significantly different—only reduced the survival of *Streptococcus mutans* by 20%. Increasing the GO concentration enhanced the antimicrobial properties of the GO coatings, but the biocompatibility of the GO-coated NiTi substrate decreased [89]. Based on the above results, the GO concentration should be controlled within a suitable range when making GO coatings, and the recommended value of this concentration has not been determined at present, while the biocompatibility of GO coatings to the human body has also not been clearly confirmed thus far. In contrast, Ag and ZnO coatings not only have reliable antimicrobial properties, but also have been applied in various medical disciplines with a higher guarantee of safety. Therefore, the earliest Ag and ZnO may still be the most suitable materials for antibacterial coatings. The main antibacterial coating materials discussed in this section are shown in Figure 5.

### 3.1. Silver and Silver Compound Coatings

Silver has a wide range of antibacterial properties, protecting the surface of the material from microbial adhesion through different mechanisms, and has a certain antibacterial effect [100,101]. AgNPs can continuously release silver ions, which cause damage to bacterial cell walls and cell membranes. They can also penetrate cells and interfere with the synthesis of proteins and deoxyribonucleic acid (DNA). Inactivation of respiratory enzymes and the production of reactive oxygen species in bacterial cells are other mechanisms that can kill the bacteria. In addition, silver nanoparticles themselves can adhere to the cell wall and the membrane surface, directly causing cell membrane denaturation and perforation, and disrupting the signal transduction of bacterial cells [99,102]. In addition to the direct killing and inhibition of bacteria, AgNPs also regulate inflammatory responses, further inhibiting the reproductive survival of microorganisms [103]. Therefore, it can be assumed that by applying a silver coating to orthodontic appliances, bacterial colonization of the dental surfaces can be reduced, reducing plaque and caries formation during orthodontic treatment.

In a study by Mhaske et al., the authors found that silver plating applied by thermal evaporation on the surface of SS and NiTi archwires reduced the adhesion of *Lactobacillus acidophilus* on the surface of archwires and showed antibacterial activity against *Lactobacillus acidophilus* [41]. AgNPs coated on SS archwire via hydrothermal synthesis also showed significant inhibition of *Staphylococcus aureus* and *Streptococcus mutans* [104].

Nanosilver-coated SS brackets also have significant antibacterial activity. Valiollah Arash et al. prepared silver particles via an electroplating method and verified the antibacterial activity of silver plating against *Streptococcus mutans* using direct contact tests and disk diffusion tests, concluding that its antibacterial activity could last for up to 30 days after application [105]. Tania Ghasemi et al. demonstrated direct inhibition of *Streptococcus mutans* reproduction by preparing nanosilver coatings on the surface of brackets using PVD [63]. In addition to its antiadhesive and antibacterial properties against *Streptococcus mutans* and *Streptococcus distortus* [106], the nanosilver coating was also effective against *Staphylococcus aureus* and *Escherichia coli* [107]. Moreover, this effect was not limited to SS brackets, but also applied to ceramic and cobalt-chromium (Co-Cr) alloy brackets [107].

In addition to in vitro tests, there have also been animal studies demonstrating the antibacterial activity of silver-coated brackets. Gamze Metin-Gürsoy et al. found that nanosilver-coated orthodontic brackets can inhibit *Streptococcus mutans* in dental plaque and produce low levels of nanosilver ion release in saliva, as well as reducing smooth-surface caries. However, they did not have any significant effect on the incidence of occlusal caries [108]. This is the only study to date in which surface-coated fixed appliances have been applied in animals’ oral cavities; thus, we need more animal experiments in further studies.

Different surface coating methods for silver have different effects on the improvement of antibacterial properties. In a clinical trial, Viktoria Meyer-Kobbe et al. first investigated the use of plasma-immersion ion implantation and deposition (PIIID) in SS brackets. They concluded that the antibacterial effect of PIIID silver-modified surfaces is as significant as that of electroplated silver layers and PVD silver coatings, reducing biofilm volume and surface coverage, and producing an even stronger bactericidal performance [109]. Coating of SS brackets with silver using PIIID has proven good antibacterial properties in clinical applications, and direct evidence is next needed to show its ability to reduce the incidence of white spots and plaque gingivitis caused by orthodontic appliances. In addition to this, TaAgB also reduces the attachment and growth of *Streptococcus mutans* on the SS surface. This is evidenced by the results observed via SEM; not only was there a reduction in *Streptococcus mutans* on the TaAgB surface, but the morphology of the bacteria was also altered, further confirming its antibacterial activity [68].

Silver compound coatings have been studied for many years. To increase the corrosion resistance of silver coatings, a hard coating of silver-platinum (Ag-Pt) alloy on stainless steel surfaces has been developed and demonstrated significant antibacterial efficacy against *Streptococcus mutans* and *Aggregatibacter actinomycetemcomitans,* as well as good biocompatibility [110]. However, this study did not use a bracket as the substrate, but rather an SS block. This design may not fully simulate the bracket when it is performing its function in the oral cavity, as the morphology and ligation of the brackets and archwires are not taken into account.

To further increase the antibacterial properties of the silver coating, some studies have applied Ag together with TiO_2_ to SS brackets. They demonstrated better antiadhesive and antibacterial properties against *Streptococcus mutans* and *Porphyromonas gingivalis* than Ag alone, helping to prevent dental caries and plaque accumulation, with satisfactory biocompatibility [111].

Both Ag and ZnO nanoparticle-coated SS brackets exhibit antibacterial effects against *Streptococcus mutans* and *Lactobacillus acidophilus*, and the antibacterial effects are maintained for at least 3 months. In addition, silver/zinc oxide nanoparticle composite-coated brackets had a stronger antibacterial effect on *Streptococcus mutans* and *Lactobacillus acidophilus* compared to silver nanoparticles and zinc oxide nanoparticles alone [112].

Other silver particle composite coatings are still being explored. A 2021 study concluded that a silver-polymer-coated archwire was not antibacterially effective. Its colony counts did not differ significantly from those of uncoated archwires in 0% sucrose and 3% sucrose environments [113]. Of course, this study only added a single species of bacteria (*Streptococcus mutans*) as the test bacteria, whereas in reality, the environment and strain composition in the oral cavity are more complex. To verify the accuracy of this result, further experiments under conditions that more closely simulate the real conditions of the oral cavity are needed.

A composite nanocoating was synthesized by Bor-Shiunn Lee et al. [114]. They applied a layer-by-layer deposition method with materials consisting of polydopamine, functionalized poly(3,4-ethylenedioxythiophene) (PEDOT), and AgNPs, and demonstrated the antifouling and antibacterial properties of this coating when applied to the surface of stainless steel materials. Its release of Ag ions does not have a harmful effect on humans, as it is biocompatible. This may be used in the future as a coating for orthodontic brackets and archwires.

Silver has been used as an antibacterial material for a long time and has also been shown to be biocompatible on numerous occasions [115]. Silver coatings are antibacterial against a number of species, including cariogenic and periodontal pathogenic bacteria, and provide a definite resistance to smooth-surface caries [108]. The long-lasting antibacterial effect allows for the use of brackets, which are an infrequently replaced component [112]. However, the shortcomings of silver coatings are their poor corrosion resistance, low hardness, and high price. The upside is that silver composite coatings with other materials have similar antibacterial effects [112], so silver composite coatings can be developed to discover potential improvements in other properties.

### 3.2. Titanium and Titanium Compound Coatings

TiO_2_, as a photosensitive material, is capable of generating hydroxyl (OH) radicals and reactive oxygen species (ROS) when irradiated by ultraviolet (UV) light, which are highly reactive when exposed to organic compounds [116,117]. Based on this principle, the antibacterial properties of TiO_2_ have been receiving attention.

Several studies have confirmed that SS and NiTi archwires coated with TiO_2_ show antiadhesive effects against *Streptococcus mutans* [118,119,120,121], as well as bactericidal effects on *Streptococcus mutans* [118,119,120,121], *Porphyromonas gingivalis* [119], *Candida albicans,* and *Enterococcus faecalis* [120]. For brackets, TiO_2_ coatings also show antiadhesive and antibacterial properties against *Streptococcus mutans* [122,123], *Candida albicans* [122], and *Lactobacillus acidophilus* [124].

Rutile, anatase, and brookite are crystal structures in which TiO_2_ exists. Anatase is formed by anodic oxidation (AO), while rutile is formed by thermal oxidation (TO). Studies have shown that anatase films applied to titanium (Ti) and titanium silver (TiAg) plates are more effective than rutile films against *Streptococcus mutans* [125]. In contrast, Roshen Daniel Baby et al. applied TiO_2_ coatings with different crystal structures to SS brackets but came to the opposite conclusion. Their results showed that both structures of TiO_2_ have antibacterial effects against *Streptococcus mutans*, but the rutile phase has a stronger bactericidal effect and more significant cytotoxicity than the anatase phase [126]. All of these considerations mean that rutile should be avoided regardless of which crystal type has the better antibacterial effect, as its cytotoxicity reduces the biocompatibility of the material.

Recently, there have been some clinical studies of TiO_2_ coatings. Keerthi Venkatesan et al. showed that TiO_2_ nanoparticle coatings on NiTi archwires have good antiadhesive and antibacterial effects against *Streptococcus mutans*. However, by the end of 1 month, 60% of the TiO_2_ coating was lost, and the roughness of the archwire was similar to that of the uncoated one [127]. Nevertheless, the adhesion of *Streptococcus mutans* to TiO_2_-coated archwires was still lower compared to uncoated archwires, which can be attributed to the antibacterial properties of the nanoparticles [127]. In addition, the TiO_2_ nanoparticle coating on the SS archwire also showed an effect against *Streptococcus mutans* in the first and third weeks [128]. Whether on NiTi or SS archwires, the TiO_2_ coating was effective in reducing the initial bacterial adhesion [127,128]. However, although the TiO_2_ coating was shown to have a direct antiadhesive and bactericidal effect, its effect on the prevention of enamel demineralization was not significant [127]. Therefore, more experiments are needed to prove its preventive effects against enamel demineralization and periodontal disease in practice.

In addition to the TiO_2_ coating applied directly to the surface of the material, the surface of Ti or titanium alloys also oxidizes to form a TiO_2_ layer, which is responsible for the coatings of titanium compound materials having certain antibacterial properties [125]. As mentioned already, Ag is also an effective antibacterial agent, so it has also been investigated to be added to Ti to prepare TiAg alloys, which it is hoped will have a stronger antibacterial effect than a Ti coating alone. For *Streptococcus mutans*, TiO_2_ coatings on TiAg alloy plates have a significantly stronger and faster antibacterial effect than TiO_2_ coatings on Ti plates, suggesting that the addition of Ag to Ti can result in a synergistic enhancement of the effect against *Streptococcus mutans* [125]. However, for *Lactobacillus acidophilus*, although both showed antibacterial activity, the TiAg-coated samples showed no difference in resistance to *Lactobacillus acidophilus* compared to Ti [129], which means that the TiAg coating did not have stronger activity against *Lactobacillus acidophilus* than the Ti coating. Nevertheless, irrespective of their antibacterial effects, neither of them exhibited cytotoxicity [129]. Therefore, both materials have the potential for application on fixed appliances, although further research is needed to determine whether the addition of Ag will give the desired antibacterial effect. Ti coatings and TiAg coatings have not yet been applied to archwires or brackets, which is a gap for the time being. In addition, apart from the clinical studies, the above studies have not taken into account the mechanical forces to which the archwires are subjected in practice. Thus, the ability of the coatings to withstand mechanical forces has not been tested.

The shortcoming of TiO_2_ is that it only produces hydroxyl radicals under UV light irradiation. When UV light is filtered and only visible light is used, the photocatalytic activity of TiO_2_ is poor [130]. On the other hand, the nitrogen-doped (N-doped) TiO_2_ thin film substrate, as a visible-light-sensitive photocatalyst, has a significant bactericidal rate under visible-light irradiation [121]. Nitrogen-doped titanium dioxide (TiO_2−x_N_y_) films coated on the surface of SS brackets using RF magnetron sputtering produced 95.19%, 91.00%, 69.44%, and 98.86% inhibition against *Streptococcus mutans*, *Lactobacillus acidophilus*, *Actinomyces viscous*, and *Candida albicans*, respectively [131]. In addition, the coating exhibited strong antibacterial properties against *Streptococcus mutans* over a 90-day timeframe. In other words, surface coating of SS brackets with N-doped TiO_2_ films inhibits *Streptococcus mutans* for up to 3 months [132]. To obtain coatings with the highest visible-light photocatalytic activity and antibacterial activity against *Lactobacillus acidophilus* and *Candida albicans*, a sputtering temperature of 300 °C for 180 min, an aigon (Ar):nitrogen (N) ratio of 30:1, and an annealing temperature set at 450 °C can be performed [133]. The preparation of TiO_2_ nanofilms using PVD has also been examined and has shown similar results on the antibacterial effect of TiO_2_, regardless of whether the coating thickness was 60 μm or 100 μm [63]. In some studies, however, the visible-light exposure time was set to 24 h [121,131]. This condition is impossible to achieve in practical applications. Therefore, simulation experiments that are closer to the clinical situation, animal experiments, or clinical trials are needed to further demonstrate the effectiveness of N-doped TiO_2_ coatings in improving the antibacterial properties of fixed appliances. As shown in a randomized controlled clinical trial by Avula Monica et al. [134], N-doped TiO_2_-coated SS brackets are effective in reducing the extent of the increase in *Streptococcus mutans* concentration when exposed to natural visible light and dental surgical light. This effect holds for up to 60 days and is better at 30 days after the placement of orthodontic appliances than at 60 days. However, even this latest related experiment did not directly detect and document enamel demineralization around the brackets or gingival inflammation. This will need to be followed up in future studies.

When TiO_2_ is applied in combination with other materials, it is found that Ag applied with TiO_2_ on SS brackets has better antiadhesive and antibacterial properties against *Streptococcus mutans* and *Porphyromonas gingivalis* [111].

Titanium nitride is also considered to have an antibacterial effect, although TiN is not as antibacterial as TiO_2_ and carbon nitride (CN_x_) [29]. However, the study by Licia Pacheco Teixeira et al. concluded that neither TiN nor titanium nitride doped with calcium phosphate (TNCP) films interfere with the attachment of *Streptococcus mutans* to the surface of the bracket, meaning that this coating has no antiadhesive effect and does not reduce the growth of *Streptococcus mutans* [135]. Moreover, the use of TiN as the surface coating may increase the surface friction [29]. In summary, the utility of TiN as a surface coating for orthodontic materials needs to be further verified.

Mian Chen et al. [136] used titanium treated with plasma-enhanced fluorine (F) and oxygen (O) mono/double chemical vapor deposition to obtain nanofunctional coatings with improved antibacterial properties and biocompatibility. The results showed that the fluorine-deposited samples can effectively exterminate *Staphylococcus aureus* with sufficient antibacterial properties, as shown in Figure 6. More importantly, the F and O double-deposited (F-O-Ti) coatings exhibited better sustained antibacterial performance than the F single-deposited coatings after 7 days of immersion in 0.9% sodium chloride (NaCl) solution. This result tells us that this type of material may be seen as an effective antibacterial coating for clinical applications.

Titanium compound coatings rely mainly on the photoreactivity of TiO_2_ for their antibacterial action [130]. The advantages include their ability to resist caries and periodontal pathogenic bacteria [111,131,136], as well as their biocompatibility [125]. TiO_2_ coatings are probably best used for short-term applications only, as they reduce initial bacterial adhesion [134] and undergo rapid peeling [127], making them unsuitable for long-term use. In addition, the increased friction of such coatings is detrimental to therapeutic efficiency, and the need for light causes inconvenience in their use [130]. Titanium has composite applications with silver [125], nitrogen [121], fluorine, and oxygen [136], which enhance its antibacterial properties, but the durability and high friction have not been addressed thus far. Currently, difficulties remain in clinical application.

### 3.3. Zinc and Zinc Compound Coatings

ZnO nanoparticles reduce biofilm formation, with low toxicity as well as good biocompatibility [137], making them suitable for biomedical applications. In a 2015 study, Baratali Ramazanzadeh et al. found that ZnO-nanoparticle-coated brackets prepared via spray pyrolysis had antibacterial effects, but they were not as effective as copper oxide (CuO) nanoparticle coatings and CuO-ZnO composite coatings, both of which reduced the number of *Streptococcus mutans* to 0 after 2 h [138]. However, the coatings were not tested for biocompatibility. If the addition of CuO leads to elevated cytotoxicity beyond the acceptable threshold for biocompatibility, it will be difficult to apply it directly in clinical applications, even if the antibacterial properties are good enough. Unlike CuO, ZnO has proven to be biocompatible [25]. Therefore, it is safe to use for a wide range of biomedical materials.

It has been shown that ZnO nanoparticle coatings on the surface of NiTi archwires have antibacterial effects against *Streptococcus mutans* [25,139], *Staphylococcus aureus*, *Streptococcus pyogenes,* and *Escherichia coli* [27]. Different coating methods can be used to obtain ZnO particles with different physicochemical properties, such as chemical precipitation, chemical vapor deposition (CVD), electrostatic spinning, polymer composite coating, and the sol-gel method. Comparisons also reveal that the smaller the particles, the larger the specific surface area of the particles, and the more the antibacterial performance is improved [139,140]. In addition, the color of the ZnO coating is white or light gold and does not tarnish in the short term, resulting in a higher aesthetic quality [25,138].

However, a study in 2021 concluded that the antibacterial effect of ZnO coatings was not significant [123]. In this study, the investigators prepared a photocatalytic ZnO coating on the surface of SS brackets via magnetron sputtering, and the results showed that the antibacterial effect of ZnO was not satisfactory. Based on these contradictory findings, more experiments are needed in order to determine the feasibility of the clinical application of ZnO nanoparticle coatings on orthodontic archwires or brackets.

In addition to the composite application of ZnO with CuO, it can also be used to form a coating with Ag nanoparticles. Noha K. Zeidan et al. [112] made a composite coating of ZnO with silver nanoparticles and applied it to SS brackets, finding that the antibacterial effect against *Streptococcus mutans* and *Lactobacillus acidophilus* exceeded that of the application of ZnO alone. Considering the antibacterial effectiveness of silver and the mechanical properties of ZnO, the combination of the two can bring the advantages of the coating to a more powerful level.

In summary, ZnO can be used as a multifunctional coating in clinical applications. It is antibacterial, friction reducing [26], and biocompatible [25], and it is safe and aesthetically pleasing to use [25,138]. On the downside, its corrosion resistance and durability for long-term application are not yet confirmed, and the antibacterial effect is still controversial. However, its application in combination with Ag nanoparticles can provide similar antibacterial performance and is more effective than ZnO alone [112]. In addition, future studies could focus on how to obtain finer ZnO nanoparticles in the preparation of coatings to enhance the antibacterial properties.

### 3.4. Other Metal and Metallic Compound Coatings

For antibacterial surface coatings prepared from other metallic materials, such as rhodium and gold, attempts have also been made on fixed appliances, although many research gaps remain.

There is no consensus on the effect of rhodium plating on the surface of archwire in terms of antibacterial properties. Some believe that the rhodium-coated NiTi archwires have increased antibacterial properties [141], while others consider that the risk of biofilm retention on rhodium-coated NiTi archwires is as high as that of uncoated NiTi archwires [113].

In 2014, a gold-plated aesthetic archwire was shown to reduce the surface roughness of SS archwires, but it caused no reduction in the adhesion of *Streptococcus mutans*. Moreover, surface free energy (SFE) may have a positive correlation with bacterial adhesion. If the SFE of the coating is the same as that of the archwire, the antiadhesive effect may not be significant, just as the SFE of gold-plated SS archwires is similar to that of SS archwires [141]. However, another study on other coatings suggested that surface roughness is positively correlated with biofilm adhesion [142]. This contradiction may be due to different experimental designs, or to the varying nature of different coatings and their antibacterial mechanisms. According to the “attachment point theory”, biofoulers (in the context of this review, bioorganisms) have more attachment points on rough surfaces [143]. This theory may partially explain why some believe that bacteria are more likely to be attached on rough surfaces. However, smooth surfaces are not necessarily more resistant to adhesion than surfaces with undulations. Some researchers, inspired by bionics, have tried to create nanofolds on the surface of the substrate. These folds are like ripples with a certain wavelength and amplitude. When these two parameters are appropriate, the adhesion of biofoulers to the surface can be reduced. That is, although the surface is not completely smooth, its antiadhesion properties are significantly better than even that of a smooth surface when the gap between the folds is slightly smaller than the size of the bacteria [144]. This suggests that preparing nanofolds of suitable wavelengths and amplitudes on the surfaces of archwires and brackets might be a possible way to resist bacterial adhesion.

In addition, other gold materials—such as 4,6-diamino-2-pyrimidinethiol-modified gold nanoparticles (AuDAPT) [145] and quaternary ammonium (QA)-modified gold nanoclusters (QA-GNCs) [146]—can be used as antibacterial coatings for invisible orthodontic appliances. AuDAPT and QA-GNC have resistance to *Porphyromonas gingivalis* [145] and *Streptococcus mutans* [146], respectively, and they have shown excellent biocompatibility, but they have not yet been investigated in fixed appliances.

### 3.5. Polymeric and Bioactive Materials

Epoxy has been clinically demonstrated to have antibacterial properties as an aesthetic coating on the surface of NiTi archwires [147]; however, this study took the plaque directly from the surface of the archwire during orthodontic treatment and counted the number of bacterial colonies in the culture, without distinguishing different strains. Two other in vitro experiments focused on *Streptococcus mutans* and showed that epoxy coating on the surface of SS or NiTi archwires reduces the adhesion of *Streptococcus mutans* and *Streptococcus distans* to the archwires in the short term [141,148]. Regarding the mechanism of this antiadhesive effect, the authors suggested that it was due to a reduction in the SFE of the archwire, rather than a change in surface roughness [141]. Conversely, Deise C. Oliveira et al. found that the amount of biofilm adhesion was significantly higher on epoxy-coated than on uncoated NiTi archwires. In addition, there was no significant difference in the number of bacterial colonies between coated and uncoated archwires [113]. This indicates that the epoxy coating does not increase the antibacterial properties of the NiTi archwire surface, and it may even decrease the antiadhesive properties. Based on these contradictory findings, multiple replicate experiments are needed in the future to determine the accuracy of the results. Efforts are also needed to make the in vitro experimental conditions more closely resemble the intraoral physiological environment, or to conduct animal and clinical experiments.

Another polymeric aesthetic coating material that has been used for a long time is polytetrafluoroethylene, also known as PTFE. It has been shown in clinical trials to reduce the amount of plaque biofilm adhesion on NiTi archwires [147] and SS brackets [149], and it is effective on every surface of the SS bracket [149]. Additionally, PTFE coatings prepared at low temperatures have better resistance to microbial adhesion than those prepared at high temperatures [95]. Still, these antibacterial and antiadhesive effects cannot be attributed to a reduction in initial bacterial adhesion [150]. However, similar to epoxy, it has been suggested that PTFE does not reduce the amount of biofilm adhesion on the surface of NiTi archwires. Interestingly, both results came from the same research [113], and only this single study reached the opposite conclusion from the previous ones. More similar studies may be needed in the future to verify the effects of PTFE and epoxy coatings on the antibacterial properties of NiTi archwire surfaces. Furthermore, the combined usage of AgNPs and PTFE through a similar method has been investigated, showing better antibacterial properties compared to applying PTFE coatings alone.

Regarding other polymeric coatings, Adauê S Oliveira et al. [151] prepared silicon oxides (SiO_x_) hydrophobic/superhydrophobic coatings on SS and ceramic bracket surfaces via a sol-gel process. Hydrophobic coatings were made using cetyltrimethoxysilane diluted in ethanol, termed HS1; superhydrophobic coatings were made using 1H,1H,2H,2H-perfluorodecyltriethoxysilane diluted in dimethyl sulfoxide, termed HS2. The results showed that both HS1 and HS2 reduced the biofilm accumulation on the surface of the bracket in 24 h, and the superhydrophobic HS2 was more effective than the hydrophobic HS1. Based on the results of this study, when applying SiO_x_ coatings to increase the antibacterial properties of the brackets, HS2 superhydrophobic coatings can be prepared to obtain the best results [151]. The oral environment is moist, so the liquid–air interface formed between the hydrophobic surface and saliva prevents bacteria from adhering and forming biofilms [152], which explains the excellent antimicrobial properties of hydrophobic and superhydrophobic coatings. The abovementioned study was published in 2015, and there have been no subsequent studies on the surface coating of orthodontic brackets or archwires that present prospects worth exploring.

MPC polymer coatings inhibit the adhesion of *Streptococcus mutans* to SS archwires [96]. CTS has been used as an antibacterial material in the resin for bonding orthodontic attachments, with the purpose of controlling enamel demineralization, but it has not been tested for its antibacterial properties as a surface coating for orthodontic attachments, which is a direction that could be developed in the future [24]. As another polymer, polyethylene glycol (PEG)-coated SS archwires exhibited the best antiadhesive properties at a molecular mass of 5000, maintaining very low bacterial adhesion even after 10 h [153].

Bioactive materials such as lysozyme are also increasingly being used in orthodontics. New lysozyme-coated orthodontic composite archwires (CAWs) exhibit antibacterial properties in artificial saliva—mainly against *Staphylococcus aureus* [154]. In addition, the corrosion resistance and biocompatibility of the lysozyme coating are relatively considerable [154]. However, other strains of bacteria in the oral cavity—such as *Streptococcus mutans*, which often causes caries, and *Porphyromonas gingivalis*, which causes periodontal disease—have not been studied. Nevertheless, lysozyme as a bioactive material provides inspiration for follow-up studies. More bioactive materials can be explored for surface modification in the surface coating of fixed appliances. However, although lysozyme coatings have been shown to have an antibacterial effect against *Staphylococcus aureus*, their effect on Gram-negative bacteria is weaker than on Gram-positive bacteria, so there is a risk of disrupting the balance of the intraoral flora when applied to fixed orthoses. Further animal experiments are needed in order to confirm their clinical value.

There are also bioactive antimicrobial coatings that are based on antiadhesion. The protein molecule bovine serum albumin (BSA) significantly reduced bacterial adhesion on the surfaces of SS, ceramic, and resin brackets and SS archwires, with a maximum decrease of more than 95% on the surfaces of brackets, probably due to the BSA-mediated reduction in surface free energy [155]. In addition, in 2020, Peng et al. developed an antimicrobial hydrogel using a combination of PEG and chitosan (CTS/PEG) [156]. Archwires coated with these bioactive materials showed a significant increase in both antiadhesive and antimicrobial properties; thus, their possible uses in dental applications are considerable.

Recent studies have found that coatings of polydopamine (PDA) and blue fluorescent hollow carbon dots (HCDs) can maintain more than 50% antimicrobial performance against *Escherichia coli* and *Streptococcus mutans* for 14 days. The fluorescent nature of HCDs may also provide clinical visualization, allowing for timely replacement of defective coatings. In addition, the combination of low drug resistance, low toxicity, and high biosafety will allow this bioactive material to go far in the future [20].

Recent studies on antibacterial coatings for orthodontic appliances are summarized in Table 2.

**Table 2 ijms-24-06919-t002:** Recent studies on antibacterial coatings for orthodontic appliances.

Ref.	Coating Materials	Coating Technique	Substrate	Study Type	Roughness	Antibacterial Effectiveness	Other Effectiveness
[109]	Ag	Galvanic, PVD, PIIID, and deposition	SS brackets	Clinical trial	R_a_ for the untreated bracket material was 0.04 µm, R_a_ for the galvanic coating was 0.12 µm, R_a_ for the PVD coating was 0.08 µm, and R_a_ for the PIIID procedure coating was 0.06 µm.	The biofilm volume per test specimen for the control was 7.24 × 10^8^ µm^3^. For the galvanically applied silver coating, the biofilm volume decreased to 2.62 × 10^7^ µm^3^, for the PVD coating to 4.44 × 10^7^ µm^3^, and the PIIID procedure to 3.82 × 10^7^ µm^3^. The reduction of the biofilm volume compared to the control was statistically significant for all surface modifications. The percentage surface coverage per test specimen was 64.40% for the unmodified control and decreased to 16.97% for the galvanic silver surface, 23.81% for the PVD coating, and 23.63% for the PIIID-modified surface.	NA
[112]	Ag (NPs)	PVD	SS brackets	In vitro (in artificial saliva)	NA	The inhibition percent of *Streptococcus mutans* and *Lactobacillus acidophilus* were 27.60% and 62.02%, respectively.	Durability (>3 months)
[106]	Ag (NPs)	NA (commercial)	NA	In vitro (in suspensions of microorganisms)	NA	The silver coating decreased the adhesion of both *Streptococcus mutans* and *Streptococcus sanguinis* to the orthodontic brackets	NA
[104]	Ag (NPs)	Hydrothermal synthesis	SS archwires	In vitro (in suspensions of *Streptococcus mutans* and *Staphylococcus aureus*)	It was not possible to observe changes in roughness after coating.	Microbial adhesion and biofilm formation of *Staphylococcus aureus* and *Streptococcus mutans* reduced.	Aesthetic effect
[107]	Ag (NPs)	Synthesized in situ	SS and ceramic brackets	In vitro (on Mueller–Hinton agar plates)	NA	The inhibitory halos obtained by the in vitro evaluation of the antibacterial effect, in terms of brackets with silver nanoparticles with *Staphylococcus aureus* and *Escherichia coli*, showed an excellent inhibition of microbial growth compared to the bracket control, with a diameter between 9 and 10 mm.	NA
[140]	Ag (NPs)	Aqueous reduction	NiTi, CuNiTi, SS archwires, and SS brackets	In vitro (in suspensions of *Streptococcus mutans*)	Roughness values were increased in SS wires (7.094 × 10^3^ + 1 nm), followed by NiTi wires (6.234 × 10^3^ + 1 nm), and the lowest roughness value for CuNiTi wires (3.116 × 10^3^ + 1 nm).	Smaller Ag nanoparticles (16.7 μg/mL) had consistently better antimicrobial inhibition effects against the *Streptococcus mutans* strain compared to larger Ag nanoparticles (66.8 μg/mL), showing significant differences between them. The coated brackets had significantly better antiadherence activity (4.3 CFU/mL for smaller particles and 5 CFU/mL for larger particles) than the uncoated brackets (356 CFU/mL). Both sizes of Ag nanoparticles had statistically good adherence inhibition of the *Streptococcus mutans* strain for all types of orthodontic wires (SS = 26.1–52.6 CFU/mL, NiTi = 15.1–49.6 CFU/mL, and CuNiTi = 89.1–287.8 CFU/mL) compared with the control groups (SS = 346.7 CFU/mL, NiTi = 342.3 CFU/mL, and CuNiT = 376.2 CFU/mL).	NA
[112]	Ag/ZnO (NPs)	PVD	SS brackets	In vitro (in artificial saliva)	NA	The inhibition percent of *Streptococcus mutans* and *Lactobacillus acidophilus* were 45.32% and 80.29%, respectively.	Durability (>3 months)
[113]	Ag + polymer	NA (commercial)	NiTi archwires	In vitro (in suspensions of microorganisms + sucrose)	NA	No significant reduction in bacterial adhesion (0% sucrose) or biofilm accumulation (3% sucrose) was found when assessed by colony counting of *Streptococcus mutans*.	NA
[157]	AgNP/PTFE	Layer-by-layer deposition	316L SS plates	In vitro (in suspensions of *Escherichia coli*)	The surface roughness increased from 59.4 ± 6.1 nm to 158.1 ± 2.7 nm (deposition time of 6 h) and 177.3 ± 5.1 nm (deposition time of 12 h), respectively.	Coatings with 6 and 12 h deposition time could inhibit by ~75% and ~90% bacterial growth over the initial 3 days, respectively. After 7 days, coatings with 6 and 12 h deposition time still exhibited significant antibacterial activity, reducing by ~40% and ~50% of bacterial growth, respectively.	Corrosion-resistant effectiveness Biocompatibility
[158]	Al_2_O_3_ (NPs)	Atomic layer deposition	316L SS	In vitro (in simulation body fluid solution)	NA	The mean colonies forming units per milliliter were reduced by 20%; the diffusion zone was ~4 mm.	Corrosion-resistant effectiveness Biocompatibility
[158]	Al_2_O_3_/TiO_2_ Multilayer	Atomic layer deposition	316L SS	In vitro (in simulation body fluid solution)	NA	The mean colonies forming units per milliliter were reduced by 40%; the diffusion zone was >6 mm.	Corrosion-resistant effectiveness
[155]	BSA	Chemical deposition	SS, ceramic, and resin brackets; SS archwires	In vitro (in suspensions of *Streptococcus mutans*)	The adsorbed BSA molecule was not uniform on the surface, thereby leading to slight surface roughness.	After integrating BSA molecules, the three kinds of brackets all showed more than 95.0% reduction in *Streptococcus mutans* adhesion (i.e., 98.3% for SS, 96.3% for ceramic, and 95.2% for resin). Compared with bare archwires, only a few bacteria (~7.5%) could be found on the BSA-coated archwires’ surface, even after incubation in bacterial suspension for 300 min; and the optimal BSA concentration was 10 mg/mL.	NA
[156]	CTS/PEG hydrogel	Combining silane chemistry and subsequent copolymerization	SS archwires	In vitro (in suspensions of *Streptococcus mutans*)	R_q_ of CTS/PEG-coated SS archwires increased from 0.26 to 1.57 nm.	This biointerface showed superior activity in early-stage adhesion inhibition (98.8%, 5 h) and displayed remarkably long-lasting colony-suppression activity (93.3%, 7 d).	Durability (>7 days) Wear resistance Biocompatibility
[29]	CN_x_	Ion beam-assisted deposition	304L SS disks	In vitro (friction test: NA; antibacterial test: in bacterial suspension)	The films slightly reduce the surface roughness parameter. R_a_ reduced from 0.181 to 0.140 μm	Bacteria density reduced from 13.002 × 10^3^ /mm^2^ to 4.030 × 10^3^ /mm^2^ (*p* < 0.05).	Friction reduction effectiveness Biocompatibility
[113]	Epoxy	NA (commercial)	NiTi archwires	In vitro (in suspensions of microorganisms + sucrose)	NA	No significant reduction in bacterial adhesion (0% sucrose) or biofilm accumulation (3% sucrose) was found when assessed by colony counting of *Streptococcus mutans*.	NA
[142]	Epoxy	NA (commercial)	NiTi archwires	In vitro (in suspensions of *Streptococcus mutans* and *Streptococcus sobrinus*)	R_a_ = 1.29 μm (higher than uncoated archwires)	Epoxy-coated wires demonstrated an increased adhesion of *Streptococcus mutans* (5.55 CFU/cm^2^) and *Streptococcus sobrinus* (4.64 CFU/cm^2^).	NA
[136]	F-O-Ti	Plasma-enhanced fluorine and oxygen mono/dual CVD	Commercially available pure titanium with 99.9% purity	In vitro (in suspensions of *Staphylococcus aureus* and artificial saliva)	A large amount of convex texture with 100–200 nm size distributes uniformly all over the surface.	The antibacterial rates were higher than 90%.	Corrosion-resistant effectiveness Durability (~7 days) Biocompatibility
[89]	GO	Silane coupling	NiTi archwires	In vitro (friction test: NA; antibacterial test: in bacterial suspension; corrosion test: artificial saliva)	The surface of the samples coated with 2 mg/mL GO concentrations was smooth with a uniformly coated area.	The bacterial CFU values for *Streptococcus mutans* were 0.77 (samples coated with 0.5 mg/mL GO concentrations), 0.40 (samples coated with 2 mg/mL GO concentrations), and 0.23 (samples coated with 5 mg/mL GO concentrations) relative to that on bare NiTi (1.00). The number of live bacteria decreased, and the number of dead bacteria increased, indicating that GO coating could effectively resist adherent bacteria. This effect had a positive correlation with GO concentration, indicating the concentration-dependent antibacterial ability of these GO-coated surfaces.	Friction reduction effectiveness Corrosion-resistant effectiveness Wear resistance Biocompatibility
[154]	Lysozyme	Liquid phase deposition	CAWs	In vitro (in suspensions of *Staphylococcus aureus* and artificial saliva)	The surface roughness increased after being coated according to two- and three-dimensional atomic force micrographs.	When coated with 20, 40, and 60 g/L lysozyme, Live/dead bacteria staining of *Staphylococcus aureus* reduced from 90% to 82%, 59%, and 61%, respectively.	Corrosion-resistant effectiveness Durability (~2 weeks) Wear resistance Biocompatibility
[134]	N-doped TiO_2_	Radio frequency magnetron sputtering	SS brackets	Clinical trial	NA	Coated archwires (38.54 and 36.84) showed greater Ct values than uncoated wires (34.71 and 31.89) at 30 d and 60 d, respectively. Greater Ct values indicate lower *Streptococcus mutans* adhesion. Therefore, coated wires demonstrated lower *Streptococcus mutans* adhesion when compared with uncoated wires.	Durability
[132]	N-doped TiO_2_	Radio frequency magnetron sputtering	SS brackets	In vitro (in suspensions of *Streptococcus mutans*)	NA	The coating decreased *Streptococcus mutans* colonies from 401.21 to 37.82 CFU/mL.	Durability (>90 days)
[114]	PEDOT/Ag	Layer-by-layer deposition	316L SS plates	In vitro (in suspensions of *Streptococcus mutans* and *Escherichia coli*)	NA	The antiadhesive and antibacterial activity against *Streptococcus mutans* and *Escherichia coli* significantly increased.	Biocompatibility
[20]	PDA and HCDs	Electrostatic adsorption	SS brackets	In vitro (in the soaking solution of the archwires)	NA	After soaking in artificial saliva for 7 and 14 days, the coatings formed by 50 μL and 100 μL of HCDs showed antibacterial rates against *Escherichia coli* and *Streptococcus mutans* that could still reach more than 50%, but the antibacterial properties of the coatings formed by 150 μL of HCDs were much weaker than those of the above two groups.	Durability (>7 days) Biocompatibility
[113]	PTFE	NA (commercial)	NiTi archwires	In vitro (in suspensions of microorganisms + sucrose)	NA	No significant reduction in bacterial adhesion (0% sucrose) or biofilm accumulation (3% sucrose) was found when assessed by colony counting of *Streptococcus mutans*.	NA
[142]	PTFE	NA (commercial)	NiTi archwires	In vitro (in suspensions of *Streptococcus mutans* and *Streptococcus sobrinus*)	R_a_ = 0.74 μm (higher than uncoated archwires)	PTFE-coated wires demonstrated an increased adhesion of *Streptococcus mutans* (4.76 CFU/cm^2^) and *Streptococcus sobrinus* (3.73 CFU/cm^2^).	NA
[157]	PTFE	NA	316L SS plates	In vitro (in suspensions of *Escherichia coli*)	The surface roughness increased from 59.4 ± 6.1 nm to 134.7 ± 3.9 nm after being coated.	The PTFE coating only demonstrated short-term antiadhesive activity, reducing by ~45% biomass adhesion on the first day as compared with 316L SS, while coatings with 6 h and 12 h deposition time inhibited by ~65% and ~80% of biomass formation.	Corrosion-resistant effectiveness Biocompatibility
[113]	Rhodium	NA (commercial)	NiTi archwires	In vitro (in suspensions of microorganisms + sucrose)	NA	No significant reduction in bacterial adhesion (0% sucrose) or biofilm accumulation (3% sucrose) was found when assessed by colony counting of *Streptococcus mutans*.	NA
[142]	Rhodium	NA (commercial)	NiTi archwires	In vitro (in suspensions of *Streptococcus mutans* and *Streptococcus sobrinus*)	R_a_ = 0.34 μm (higher than uncoated archwires)	Rhodium-coated wires demonstrated an increased adhesion of *Streptococcus mutans* (3.85 CFU/cm^2^) and *Streptococcus sobrinus* (2.79 CFU/cm^2^).	NA
[68]	TaAgB	Magnetron sputtering with Ag-doping	304 SS sheets	In vitro (in artificial saliva)	NA	The amount of *Streptococcus mutans* adhesion on the TaAgB surface was significantly reduced and the morphology changed according to the SEM image.	Friction reduction effectiveness Biocompatibility
[29]	TiN	Ion beam-assisted deposition	304L SS disks	In vitro (friction test: NA; antibacterial test: in bacterial suspension)	The films slightly reduce the surface roughness parameter. R_a_ reduced from 0.181 to 0.162 μm.	Bacteria density reduced from 13.002 × 10^3^ /mm^2^ to 3.888 × 10^3^ /mm^2^ (*p* < 0.05).	Friction reduction effectiveness Biocompatibility
[135]	TiN	Cathodic cage deposition	SS brackets	In vitro (in suspensions of *Streptococcus mutans*)	NA	The presence of coatings did not influence the formation of the *Streptococcus mutans* biofilm (*p* = 0.06).	NA
[29]	TiO_2_	Ion beam-assisted deposition	304L SS disks	In vitro (friction test: NA; antibacterial test: in bacterial suspension)	NA	Bacteria density reduced from 13.002 × 10^3^ /mm^2^ to 1.368 × 10^3^ /mm^2^ (*p* < 0.05).	Wear resistance Biocompatibility
[123]	TiO_2_	Radio frequency magnetron sputtering	SS brackets	In vitro (in suspensions of *Streptococcus mutans*)	NA	The “weight increase” reduced from 1.0100 × 10^3^ mg to 0.6750 × 10^3^ mg, which showed increased antiadhesive activity against *Streptococcus mutans*. The survival rate reduced from 3548.3350 to 2895.0000 CFU/mL, which showed increased antibacterial activity against *Streptococcus mutans*.	NA
[127]	TiO_2_	Radio frequency magnetron sputtering	NiTi archwires	Clinical trial	R_a_ reduced from 1591.08 to 746.14 nm.	Coated archwires (30.97) showed greater Ct values than uncoated wires (37.00). Greater Ct values indicate lower *Streptococcus mutans* adhesion. Therefore, coated wires demonstrated lower *Streptococcus mutans* adhesion when compared with uncoated wires. This difference was statistically significant (*p* = 0.0005).	Durability (~1 month)
[128]	TiO_2_	PVD	SS archwires	Clinical trial	NA	The coating decreased *Streptococcus mutans* colonies from 5122.0 to 1400.0 and from 1141.8 to 297.7 CFU/mL in the first and the third week, respectively.	Durability (>3 weeks) Biocompatibility
[158]	TiO_2_ (NPs)	Atomic layer deposition	316L SS	In vitro (in simulation body fluid solution)	NA	The mean colonies forming units per milliliter were reduced by 15%; the diffusion zone was ~2 mm.	Corrosion-resistant effectiveness Biocompatibility
[135]	TNCP	Cathodic cage deposition	SS brackets	In vitro (in suspensions of *Streptococcus mutans*)	NA	The presence of coatings did not influence the formation of the *Streptococcus mutans* biofilm (*p* = 0.06).	NA
[123]	ZnO	Radio frequency magnetron sputtering	SS brackets	In vitro (in suspensions of *Streptococcus mutans*)	NA	The antiadhesive and antibacterial activity against *Streptococcus mutans* showed no significant increase.	NA
[112]	ZnO (NPs)	PVD	SS brackets	In vitro (in artificial saliva)	NA	The inhibition percent of *Streptococcus mutans* and *Lactobacillus acidophilus* were 17.54% and 28.85%, respectively.	Durability (>3 months)
[139]	ZnO (NPs)	CVD, chemical precipitation method, polymer composite coating, sol-gel synthesis, the electrospinning process	NiTi archwires	In vitro (in suspensions of *Streptococcus mutans*)	The sizes of the NPs were 59–61 nm, 30–150 nm, and 28 nm when coated during CVD, chemical precipitation method, and sol-gel synthesis process, respectively. Electrospinning gave a branch of fibers gathered together in a network, and polymer composite coating showed a nonuniform and excursive surface.	The highest *Streptococcus mutans* antibacterial effect with 98%, 96%, and 93% microbial cell reduction belonged to CVD, precipitation method, and sol-gel synthesis, respectively, and the lowest cell reduction was seen in the electrospinning method (72%)	NA
[27]	ZnO (NPs)	Electrochemical deposition	NiTi archwires	In vitro (friction test: NA; antibacterial test: on nutrient agar plates)	NA	The inhibition zone of *Staphylococcus aureus*, *Streptococcus pyogenes*, and *Escherichia coli* was formed around all ZnO nanoparticles coated archwires, with diameters of 4.25, 6.25, and 3.57 mm, respectively.	Friction reduction effectiveness
[159]	ZrO_2_	EBPVD	316L SS	In vitro (in artificial saliva and artificial saliva containing NaF)	R_a_ reduced from 10 to 3 nm after being coated.	The bacterial adhesion reduced from 8 to 5.5 × 10^3^ CFU/cm^2^.	Corrosion-resistant effectiveness Wear resistance Biocompatibility

NA: not applicable; PIIID: plasma immersion ion implantation; CuNiTi: copper-nickel-titanium; CFU: colony-forming unit; Ag/ZnO: silver/zinc oxide; AgNP/PTFE: nanosilver/polytetrafluoroethylene; Al_2_O_3_: aluminum oxide; Al_2_O_3_/TiO_2_: aluminum oxide/titanium dioxide; BSA: bovine serum albumin; CTS/PEG: chitosan/polyethylene glycol; F-O-Ti: fluorine and oxygen double-deposited titanium; GO: graphene oxide; CAWs: orthodontic composite arch wires; Ct: cycle threshold; N: nitrogen; PEDOT/Ag: poly(3,4-ethylenedioxythiophene)/silver; PDA: polydopamine; HCDs: blue fluorescent carbon dots; TNCP: titanium nitride doped with calcium phosphate; CVD: chemical vapor deposition; ZrO_2_: zirconium oxide; EBPVD: electron beam-physical vapor deposition.

## 4. Corrosion-Resistant Coatings

Research on corrosion-resistant coatings has focused on nitride- and carbonized-titanium compound materials, as well as diamond-like material coatings. Although other carbon-based materials such as graphene are also available, their effectiveness is still not clearly recognized. In particular, a common phenomenon in corrosion-resistant coatings is that their durability has not been confirmed by long-term experiments. Other properties of aesthetic coated polymers such as epoxy and PTFE have recently been investigated in the literature, demonstrating their favorable corrosion resistance for orthodontic archwires. This versatility gives more space for the development of polymeric aesthetic coatings. The main corrosion-resistant coating materials discussed in this section are shown in Figure 7.

### 4.1. Titanium and Titanium Compound Coatings

NiTi alloys have a tendency to corrode in chlorine- and fluorine-containing solutions [160], as does stainless steel. Some surface coatings can enhance their corrosion resistance, but coatings might wear and peel off after a period of application, resulting in increased roughness and reduced aesthetics [161]. Therefore, the surface coating of fixed appliances needs to be resistant to wear as well as corrosion in order to provide better durability.

As early as 2002, a study concluded that the corrosion resistance of ion-plated titanium nitride (TiN) SS brackets in artificial saliva was not significantly better than that of uncoated SS brackets [162]. Later, the opposite conclusion was reached, suggesting that TiN-plated SS brackets had increased corrosion resistance [28] and wear resistance [163] in artificial saliva. However, at that time, the corrosion resistance of TiN coatings in fluorine-containing solutions had not been confirmed in studies [28]. It was not until Xue-shun Yuan et al. prepared N-doped TiO_2_ films on SS brackets via radio frequency magnetron sputtering that they were demonstrated to increase the corrosion resistance of SS brackets in both artificial saliva and 1.23% acidulated phosphate fluoride (APF) solution [164]. In addition to brackets, TiN-coated SS archwires have shown good corrosion resistance in both acidic saliva [19] and electrochemical corrosion experiments [58].

Recently, it has been found that TiN coatings have favorable wear resistance but are more likely to fracture due to bending [165]. However, for coatings on SS substrates, some new materials also have higher corrosion resistance than TiN. These coatings include titanium-diamond-like carbon (Ti-DLC) [166] and TiCN [31], and the corrosion resistance of TiCN increases with the nitrogen content and carbon content used in the preparation process [31]. In addition to SS, the use of TiCN coatings also reduced the amount of ions released from nickel-chromium-molybdenum (Ni-Cr-Mo) alloy substrates [167] and the mass loss of nickel-chromium (Ni-Cr) alloy substrates [168], indicating that the coatings improved the corrosion resistance of various metallic materials.

In addition to carbon, aluminum compounded with nitrogen and titanium also has a good effect. TiAlN coatings applied by cathodic arc physical vapor deposition (CA-PVD) are also capable of reducing the corrosive effect of fluoride on TMA wire and are more effective than tungsten carbide/carbon (WC/C) coatings [59].

For NiTi brackets, multilayer TiN/Ti coatings deposited by pulsed bias arc ion plating (PBAIP) have higher corrosion resistance than a single layer of TiN in artificial saliva [169]. For NiTi archwires, TiN coatings formed by nitriding [170,171] or electroplating [19] reduce the precipitation of metal ions, and ion-plated TiN coatings show a visible reduction in the area of corrosion spots on the surface of the archwire [58]. TiN/Ti coatings formed by PVD not only improve the corrosion resistance of the archwire, but also its resistance to friction loss [172]. Liu, Jia-Kuang et al. [172] further discovered that in TiN/Ti coatings, the TiN layer provides protection against mechanical damage, while the Ti layer improves the corrosion resistance. In multilayer TiN/Ti, the improvement of wear resistance is similar for a single layer, two layers, and four layers, and when the number of layers continues to increase, the wear resistance of the substrate is reduced instead [33]. Therefore, it is recommended to use multilayer TiN/Ti coatings with four or fewer layers to enhance wear resistance and corrosion resistance.

It has been shown that the corrosion resistance of composite archwires (CAWs) can be significantly improved by coating them with TiO_2_ nanocrystalline films—especially by using N-doped TiO_2_ nanocrystalline films [121]. However, a study by Kielan-Grabowska Z et al. [173] showed that TiO_2_- and Ag-doped TiO_2_-coated SS materials have reduced corrosion resistance. In addition, although Al_2_O_3_ coatings are rarely used alone for orthodontic brackets and archwires, they can be coated together with TiO_2_. Muna Khethier Abbass et al. prepared nanocoated films of Al_2_O_3_, TiO_2_, and multilayer aluminum oxide/titanium dioxide (Al_2_O_3_/TiO_2_) on SS surfaces via atomic layer deposition (ALD) and found that all three films could effectively enhance corrosion resistance, and that multiple layers are better than a single layer, while Al_2_O_3_ is better than TiO_2_ [158]. However, in a recent study, researchers formed nonporous, nickel-free TiO_2_ coatings with a thickness of 50 nm via pulse anodization, as shown in Figure 8, which exhibited good hydrophilicity and corrosion protection [174]. To date, there has been no study comparing the differences in the performance of this nonporous nickel-free TiO_2_ coating with compared with N-doped TiO_2_ film and Al_2_O_3_/TiO_2_ coating, which would be necessary for the consideration of practical applications.

In addition to titanium nitrides and oxides, metallic titanium and other composite coatings are also under research. Firstly, titanium sputter-coated NiTi archwire has good adhesion and corrosion resistance even after 30 days of exposure to artificial saliva [175]. Secondly, titanium-chromium-nitrogen (Ti-Cr-N) coatings obtained via radio frequency reactive sputtering deposition annealed at 400 and 700 °C can reduce the chromium released from SS substrates by ~67% in artificial saliva, and the effect is better at 400 °C than at 700 °C [176]. Another example is that titanium-niobium (Ti-Nb) coatings prepared via laser melting methods on NiTi substrates reduce the release of nickel ions [177]. These are surface coatings that can potentially be applied to orthodontic fixed appliances to prevent wear and metal sensitization of orthodontic attachments.

As TiO_2_ is not the most optimal material for corrosion-resistant coatings, TiN is relatively more worthy of research [158,173]. TiN coatings have the advantage of being effective in enhancing the corrosion resistance of brackets and archwires in both chloride- and fluoride-containing solutions [164], but they also have disadvantages, such as the tendency to fracture when bending and the potential adverse effect on friction [165]. Based on these conditions, TiN can be used as a corrosion-resistant coating for fixed appliances in patients who normally use mouthwash. In addition, carbon or other metallic materials can be added to the material, or a double-layer titanium/titanium nitride (Ti/TiN) coating can be prepared to compensate for the deficiencies of TiN applied alone [33]. In terms of coating method, a coating method that can be applied chairside is preferable, so that the archwire can bend first before coating to prevent undesirable coating peeling and archwire corrosion.

### 4.2. Carbon-Based Coatings

Around 15 years ago, S. Kobayashi et al. [76] and Yasuharu Ohgoe et al. [178] used ion beam plating to prepare DLC films on the surface of NiTi archwires. They found that the DLC coating was able to reduce the release of nickel ions in physiological saline at 80–85 °C by as much as 80% after 5 days of immersion [178], while still remaining corrosion resistant after 14 days [76]. Additionally, the DLC-coated NiTi archwire still reduced the nickel ion concentration in the solution by 16.7% after 6 months of immersion in physiological saline at 37 °C [179]. DLC coating not only enhances the corrosion resistance of NiTi archwire in warm saline, but also alleviates the toxic effects of nickel ion release on cells, and shows good biocompatibility [178,179]. It also has excellent friction resistance and will not peel off the surface of the archwire easily [76]. Similarly, on the SS filament surface, DLC coatings can be deposited to provide comparable friction reduction and corrosion resistance [82].

The corrosion of metals by fluorine ions should not be neglected either. Therefore, a study investigated the performance of MCECR plasma sputtering coatings in high-fluoride-ion environments. The results indicated that the DLC coating of the NiTi archwire under a high-fluorine environment reduced the change in the surface roughness of the archwire due to corrosion by 91.3% [83], verifying the outstanding corrosion resistance of DLC coatings. DLC-coated NiTi archwire prepared by the same method has also been shown to have strong friction resistance, significantly reducing the micromotion wear of the archwire [80]. There is still room for improvement of this excellent performance of DLC coatings. For example, deep cryogenic treatment (DCT) at temperatures from −120 to −196 °C can improve adhesion strength from 30.08 N to up to 40.54 N [180]. Therefore, this post-treatment method may be a potential way to improve wear resistance.

Graphene is valued for its excellent mechanical properties in dental material applications. Both graphene oxide (GO) and GO/Ag coatings prepared by Viritpon Srimaneepong et al. using electrophoretic deposition (EPD) on the surface of NiTi alloys had lower corrosion rates than uncoated NiTi alloys, indicating that they can enhance the corrosion resistance of NiTi alloys, as shown in Figure 9 [181]. GO can also improve the wear resistance of NiTi archwires. Dai, Danni et al. found that many grooves caused by wear were produced on the surface of uncoated NiTi archwire, but the width of grooves on the surface of GO-coated archwire was less [89]. In addition, a GO coating concentration of 2 mg/mL resulted in less wear and a smoother surface after mechanical friction compared to GO coatings at concentrations of 0.5 or 5 mg/mL [89]. This indicates that coating with GO at a concentration of 2 mg/mL results in better wear resistance. Therefore, this is the reference value when GO coatings are applied to the surface of NiTi archwires.

GSEC is another form of graphene as a surface coating, which also has good wear resistance. As observed by Pan et al. [90], GSEC-coated SS archwire produces significantly fewer wear marks than bare SS archwire after being subjected to friction [90], while the wear rate can reach a minimum of 0.11 × 10^−6^ mm^3^/Nm [46].

Fróis, António et al. coated SS brackets and archwires with hydrogenated forms of diamond-like carbon (aC:H) sputter coating via the magnetron sputtering method. They demonstrated its chemical and mechanical inertness and ability to reduce the pitting corrosion that occurs on SS substrates. However, the leaching of metal ions in the aC:H coating was higher than that in the uncoated SS [182]. What should be considered in this case, if it is to be applied, is the need to reduce either the damage to the surface morphology or the metal allergy due to the leaching of metal ions. However, studies on this coating are still insufficient, and more research should be carried out in the future.

Among the carbon-based coatings, DLC, GO, and GSEC have similar corrosion resistance advantages. However, DLC shows the best clinical application prospects. The DLC coating is not only durable [179], but also allows fixed aligners to withstand certain thermal and fluoridated environments [83]. Specifically, it reduces the corrosion that occurs on the archwire and brackets under the influence of a hotter diet or fluoride mouthwash. DLC coatings, in turn, have the ability to reduce friction [80] and are more biocompatible [178,179]. Although the application of DLC coatings leads to a decrease in the surface hardness of the attachments, this can be enhanced and the corrosion resistance further increased by means of DCT [180].

### 4.3. Other Metal and Metallic Compound Coatings

Some other metal and metallic compound coatings have corrosion resistance, but are mainly used for friction reduction, such as WS_2_ [43], ZnO [24,25,26], and TiN [29,31,32]. The potential of the remaining metallic materials as corrosion-resistant coatings is still not enough for clinical applications, so there is still space for corrosion-resistant coatings to be developed using metallic materials. Among all metal coatings, only TaAgB has good performance in friction reduction, antibacterial activity, and corrosion resistance, as a multifunctional coating with good application prospects [68].

IF-WS_2_ has been applied for a long time to reduce the friction on the surface of archwires and has been proven to have good wear resistance. In a study by Redlich, Meir et al. [183], after 100 cycles under dry conditions, very little surface wear was observed on the Ni + IF-WS_2_-coated SS archwire, and no peeling of the coating was observed—as opposed to the uncoated or Ni-coated archwires, which were severely worn after only 10 cycles [183]. This indicates that WS_2_ is highly resistant to wear and tear. In recent years, Antonio Gracco et al. have come to similar conclusions and have further verified that WS_2_ has good resistance to wear even under wet conditions [18]. Additionally, tungsten disulfide (MoS_2_) has similar properties to WS_2_ [18]. Both of them have good performance in reducing friction and wear. Therefore, the use of either MoS_2_ or WS_2_ as the surface coating of SS archwires can maintain the durability of their effects.

Solid TaAgB solution coating on SS blocks have high hardness, smoothness, and wear resistance, with a wear rate of only 6.51 × 10^−15^m^3^/Nm [68]. Based on its low-friction and antibacterial properties described above, it could be applied in the future as a multifunctional coating for orthodontic fixed appliances.

Rhodium is commonly used as an aesthetic coating and, in addition to its aesthetic properties, also has the property of reducing friction on the surface of the archwire, as well as potential antibacterial properties. However, there is more distrust in the corrosion resistance of rhodium. Katić Višnja et al. [170] and Milena Carolina de Amorim et al. [184] both suggested that rhodium coating reduces the corrosion resistance and electrochemical resistance of NiTi archwires in artificial saliva, making them more susceptible to localized pitting corrosion on the surface. Lina M. Escobar et al. [185] described rhodium-plated NiTi archwire as having a significantly rougher surface and many scratches on its surface after testing. This indicates that the wear resistance of rhodium plating is also less than satisfactory.

Although copper-coated NiTi archwire has sufficient corrosion resistance [186], the biocompatibility of copper and copper oxide coatings is yet to be confirmed, due to the cytotoxicity of copper ions, among other characteristics. Until then, its use in clinical settings requires caution. Gold-plated SS archwire is resistant to friction damage, although it is not durable under bending [165]. The performance characteristics of these materials have not been completely evaluated, and more research is needed.

Zinc oxide [187], boron-doped hydroxyapatite (B-HAp) [188], and composites of both materials [189] have been well studied as surface coatings for titanium alloys. Their resistance to corrosion, wear and tear, and peeling has been demonstrated. Zirconium oxide and zirconium oxide-silica composite coatings have also been shown to have excellent corrosion resistance. The former was used to coat a stainless steel metal block surface using the electron-beam physical vapor deposition (EBPVD) technique [159], while the latter was deposited on a titanium metal block surface using the sol-gel method [190]. However, the above materials have not been verified in terms of corrosion and wear resistance in orthodontic archwires and brackets, but are more focused on friction reduction and antibacterial properties. More research should be conducted in the future on their performance in terms of wear resistance and corrosion resistance.

### 4.4. Polymeric and Bioactive Coatings

Epoxy resin has been proven to be effective in improving the corrosion resistance of NiTi archwires in artificial saliva [191], exhibiting a significant decrease in their electrochemical corrosion tendency [184,191] and nickel ion release [184,186]. Furthermore, its ability to reduce the release of nickel ions has been confirmed in a double-blind randomized clinical trial [81].

The corrosion resistance of PTFE is stronger than that of epoxy resin [191]. The corrosion rate of a PTFE-coated NiTi archwire was 10 times lower than that of uncoated NiTi substrates [192]. However, PTFE may exhibit slight cytotoxicity to fibroblasts—approximately 36% [185]. Therefore, careful consideration is needed when choosing this material. If a high level of biocompatibility is required, it is typically better to choose epoxy resins. Recently, Zhang, Shuai et al. immobilized PTFE nanoparticles in a sol-gel matrix and dip-coated them onto 316L SS via a mussel-inspired method, followed by AgNPs deposition. The nanosilver/polytetrafluoroethylene (AgNP/PTFE) coating obtained in this way can be used on metal implant surfaces because of its good antibacterial and anticorrosion properties [157]. This coating has not been applied in orthodontics thus far and could be subjected to trials.

Polyethylene naphthalate (PEN)-coated SS archwires exhibit virtually no chromium, manganese, iron, or nickel ion leaching in HCl compared to uncoated archwires, and they consistently provide significant corrosion resistance over a bending range of 0–120° [165]. It can be assumed that the PEN coating on the SS surface has good corrosion resistance and durability to withstand large-angle bending of the archwire.

The research applications of superhydrophobic coatings are also on the rise. Cheng-Wei Lin et al. prepared a double-layer grid-blasted plasma-polymerized (GB-PP) superhydrophobic coating on an SS surface. Even after a period of toothbrush cleaning action or food chewing, the water contact angle (WCA) still exceeded 90°, indicating that the hydrophobic properties are always maintained. Moreover, its surface morphology and microstructure were still similar to those at the beginning of the coating, indicating its good durability [193].

There are a few bioactive materials used in orthodontic appliances, such as lysozyme. Longwen He et al. [154] applied a lysozyme coating to a CAW surface via liquid-phase deposition and examined its multiple properties. After 2 weeks of immersion in artificial saliva at 37 °C, the substrate with 40 g/L lysozyme showed the least corrosion pitting, cracking, and roughness. Therefore, the best concentration to use as a corrosion- and wear-resistant coating for CAW surfaces may be 40 g/L. However, no similar studies have been conducted for common SS and NiTi archwires and brackets. In the future, we may try to widen the application range of such coatings.

Studies on corrosion- and friction-resistant polymers and bioactive surface coatings are relatively new and few in number. Epoxy coatings have better corrosion resistance and friction reduction than PTFE [191], but they have not been compared to other polymeric coatings. As no obvious defects have been found so far, epoxy is one of the most promising multifunctional coatings while the performance of other polymeric coatings is not conclusive in all respects. However, the various studied polymeric coatings have good friction and corrosion resistance, which may lead to them becoming a research trend in the future.

### 4.5. Ion Injection Coatings

In 2021, Rasha A. Ahmed et al. [194] formed an ionic liquid (IL) coating on the surface of a kind of nickel-titanium-cobalt shape memory alloy (Ni_47_Ti_49_Co_4_) archwire and demonstrated the excellent corrosion resistance of the coating through electrochemical experiments and 0.1% sodium fluoride (NaF) immersion experiments. Moreover, the addition of 0.015–0.05% albumin to any IL concentration could further improve the corrosion resistance of the coated surface through the deposition of albumin on the surface of coated or uncoated archwires. However, this is the only study on the corrosion and wear resistance of IL coatings, and it is also a relatively recent development. This field is still in a nascent state and deserves more exploration.

Recent studies on corrosion-resistant coatings for orthodontic appliances are summarized in Table 3.

**Table 3 ijms-24-06919-t003:** Recent studies on corrosion-resistant coatings for orthodontic appliances.

Ref.	Coating Materials	Coating Technique	Substrate	Study Type	Roughness	Corrosion Resistant Effectiveness	Other Effectiveness
[182]	aC:H	Reactive magnetron sputtering	SS brackets, tubes, and bands	In vitro (in Fusayama-Meyer artificial saliva)	The coating presented a low R_a_ of ~7 nm.	Right after 7 days of immersion, the Cr release rate from coated samples was ~40% higher than that from uncoated samples, but was similar at day 30. Ni release from coatings was ~10% lower than uncoated samples after 7 days but ~55% higher after 30 days. Finally, Fe release was ~15% lower and similar to uncoated samples after 7 and 30 days, respectively. However, there were no segregation, metallic inclusion, delamination, or detachments on coated samples after 30 days of immersion.	Wear resistance Durability (>30 days)
[157]	AgNP/PTFE	Layer-by-layer deposition	316L SS plates	In vitro (in suspensions of *Escherichia coli*)	The surface roughness increased from 59.4 ± 6.1 nm to 158.1 ± 2.7 nm (deposition time of 6 h) and 177.3 ± 5.1 nm (deposition time of 12 h).	The I_corr_ values reduced from 2.01 × 10^−6^ to 1.58–2.51 × 10^−7^ A/cm^2^, indicating enhanced corrosion protection. However, the corrosion protection was still lower than the PTFE coatings.	Antibacterial effectiveness Durability (>7 days) Biocompatibility
[158]	Al_2_O_3_ (NPs)	Atomic layer deposition	316L SS	In vitro (in simulation body fluid solution)	NA	The corrosion rate calculated by current density was 1.913 mpy for an uncoated sample and reduced after coated to 0.203 mpy for titania film, 0.174 mpy for alumina film, and 0.164 mpy for multilayer. The corrosion resistance was effectively enhanced by thin films, multilayer proved to be more corrosion protection than single layers, and Al_2_O_3_ had better corrosion resistance than TiO_2_.	Antibacterial effectiveness Biocompatibility
[158]	Al_2_O_3_/TiO_2_ Multilayer	Atomic layer deposition	316L SS	In vitro (in simulation body fluid solution)	NA	The corrosion rate calculated by current density was 1.913 mpy for an uncoated sample and reduced after coated to 0.203 mpy for titania film, 0.174 mpy for alumina film, and 0.164 mpy for multilayer. The corrosion resistance was effectively enhanced by thin films, multilayer proved to be more corrosion protection than single layers, and Al_2_O_3_ had better corrosion resistance than TiO_2_.	Antibacterial effectiveness
[70]	Al-SiO_2_	Magnetron sputtering	NiTi and SS archwires	In vitro (in artificial saliva)	NA	The I_corr_ values decreased from 23.72 to 1.21 μA/cm^2^ and from 0.22 to 0.06 μA/cm^2^ after coating with Al-SiO_2_ on the NiTi archwires and SS archwires, respectively.	Friction reduction effectiveness Wear resistance Biocompatibility
[188]	B-HAp	Electrophoretic deposition	NiTi alloy	In vitro (in simulated body fluid)	NA	The corrosion rate of 10 wt.% B-HAp and 15 wt.% B-Hap coatings reduced from 0.055 to 0.046 and 0.036 mpy, respectively, which showed better corrosion-resistant effectiveness than other concentrations and only-HAp coatings.	Adhesion strength (up to 20–30 Mpa)
[184]	Epoxy	NA (commercial)	NiTi archwires	In vitro (in artificial saliva)	NA	The Ni release was reduced from 8.36 to 0.57 mg/L.	Durability (>30 days)
[136]	F-O-Ti	Plasma-enhanced fluorine and oxygen mono/dual chemical vapor deposition	Commercially available pure titanium with 99.9% purity	In vitro (in suspensions of *Staphylococcus aureus* and artificial saliva)	A large amount of convex texture with 100–200 nm size distributes uniformly all over the surface.	I_corr_ reduced from 0.22 to 0.09 μA/cm^2^	Antibacterial effectiveness Durability (~7 days) Biocompatibility
[193]	GB-PP	Pulsed-direct current plasma-enhanced chemical vapor deposition	AISI 304 SS	In vitro (in modified Fusayama artificial saliva)	R_a_ increased from 0.062 to 10.64 μm after being coated.	The coating particles were retained on surfaces even after being worn 500 times by a toothbrush, peanut, and nougat, while the uncoated substrates showed some scratching and surface pitting or cavity after the wear tests, especially by peanut and nougat.	Wear resistance Durability
[89]	GO	Silane coupling	NiTi archwires	In vitro (friction test: NA; antibacterial test: in bacterial suspension; corrosion test: artificial saliva)	The surface of the samples coated with 2 mg/mL GO concentrations was smooth with a uniformly coated area.	When coated with 0.5, 2, or 5 mg/mL GO concentrations, nickel ion release decreased from 20.75 to 19.75, 18.00, and 17.75 μg/L cm^2^, weight loss decreased from 0.31% to 0.29%, 0.25%, and 0.23%, I_corr_ decreased from 0.696 to 0.547, 0.381 and 0.504 μA/cm^2^, upon application of 4 mm dislocation, respectively.	Friction reduction effectiveness Antibacterial effectiveness Wear resistance Biocompatibility
[181]	GO	Electrophoretic deposition	NiTi alloy	In vitro (in 3.5% NaCl solution)	R_a_ increased from 7.55 to 11.67 nm after being coated.	I_corr_ reduced from 0.158 to 0.017 μA/cm^2^, and E_corr_ increased from −0.170 to 0.031 V vs. SCE, which exhibited good corrosion resistance.	Biocompatibility
[181]	GO/Ag	Electrophoretic deposition	NiTi alloy	In vitro (in 3.5% NaCl solution)	R_a_ increased from 7.55 to 18.43 nm after being coated.	I_corr_ reduced from 0.158 to 0.002 μA/cm^2^, and E_corr_ increased from −0.170 to 0.008 V vs. SCE, which exhibited good corrosion resistance.	Biocompatibility
[154]	Lysozyme	Liquid phase deposition	CAWs	In vitro (in suspensions of Staphylococcus aureus and artificial saliva)	The surface roughness increased after being coated according to two- and three-dimensional atomic force micrographs.	When coated with 20, 40, and 60 g/L lysozyme, the copper ion release of the archwires reduced from 0.225 to 0.20, 0.1,5, and 0.125 μg, respectively.	Antibacterial effectiveness Wear resistance Durability (>2 weeks) Biocompatibility
[174]	Ni-free oxide layer	Pulsed anodization	NiTi plate	In vitro (in PBS solution)	R_a_ of the anodized surfaces, calculated from the SPM image, was 1.78 nm for the current-anodized surface and 1.16 nm for the pulse-anodized surface. The current-anodized surface included tiny pores (~10 nm), where the depth was determined as ~10 nm; these pores did not appear on the pulse-anodized surface.	The pores on the surface reduced, and Ni release of a pulse-anodized surface at 168 h reduced from ~0.55 to ~0.2 μg·cm^−2^; however, Ni release of a current-anodized surface at 168 h increased from ~0.55 to ~1.0 μg·cm^−2^.	Durability (>168 h) Biocompatibility
[57]	Ni-Ti-Cr	Chronopotentiometry	NiTi archwires	In vitro (in artificial saliva)	NA	E_corr_ of electrodes of coated archwires (−427–328 mV) were higher than the uncoated archwires (−447 mV), which represented the corrosion-resistant effectiveness of the coatings.	Friction reduction effectiveness Durability (>60 days)
[57]	Ni-Ti-Mo	Chronopotentiometry	NiTi archwires	In vitro (in artificial saliva)	NA	E_corr_ of electrodes of coated archwires (−395−366 mV) were higher than the uncoated archwires (−447 mV), which represented the corrosion-resistant effectiveness of the coatings.	Friction reduction effectiveness Durability (>60 days)
[157]	PTFE	NA	316L SS plates	In vitro (in suspensions of *Escherichia coli*)	The surface roughness increased from 59.4 ± 6.1 nm to 134.7 ± 3.9 nm after being coated.	The PTFE coating exhibited the best substrate protection as the I_corr_ parameter was over one order of magnitude lower in value than the 316L SS substrate.	Antibacterial effectiveness Biocompatibility
[185]	PTFE	NA (commercial)	NiTi archwires	In vitro (using before-use and after-use samples)	NA	A significant reduction (17.2%) in the percentage of Ni was observed between before- and after-use archwires, which exhibited an unsatisfactory corrosion resistance.	Wear resistance Durability (>2 months) Biocompatibility
[185]	Rhodium	NA (commercial)	NiTi archwires	In vitro (using before-use and after-use samples)	NA	A significant reduction (9.6%) in the percentage of Ni was observed between before- and after-use archwires, which exhibited an unsatisfactory corrosion resistance.	Wear resistance Durability (>2 months) Biocompatibility
[184]	Rhodium	NA (commercial)	NiTi archwires	In vitro (in artificial saliva)	NA	The Ni release was reduced from 8.36 to 1.52 mg/L.	Durability (>30 days)
[31]	TiCN	Direct current reactive magnetron sputtering	316L SS and (100)-oriented Si substrates	In vitro (in artificial saliva)	NA	The corrosion resistance of TiCN increased with the nitrogen content and carbon content of the preparation process when the N_2_ flux was larger than 2.5 sccm.	Friction reduction effectiveness Aesthetic effect
[166]	TiCN	Multi-arc ion plating	316L SS plates	In vitro (in artificial saliva)	NA	The TiCN film showed a corrosion current density of ~7 μA/cm^2^, which showed better corrosion resistance than TiN, but not as good as Ti-DLC.	NA
[167]	TiCN	Magnetron sputtering	Ni-Cr-Mo disks	In vitro (dry condition)	R_a_ increased from 0.33 to 0.38–0.45 μm after being coated.	There were no visible areas with an elevated content of nickel or chromium, which proved that the base had not been revealed.	Wear resistance
[168]	TiCN	Magnetron sputtering	Ni-Cr alloy	In vitro (in neutral salt spray and seawater acetic acid)	NA	The mass loss of the examined samples reduced from 0.37 to ~0.20 mg/mm^2^ × 10^−4^.	Durability (>30 days)
[176]	Ti-Cr-N	Radiofrequency reactive sputtering	304 SS	In vitro (in artificial saliva)	The RMS roughness increased from 5 to 105–20 nm after being coated.	Ti-Cr-N coatings reduced 304 SS’s release of chromium species by ~67%, and annealed at 400 °C displayed higher corrosion resistance.	Durability (>90 days)
[166]	Ti-DLC	Multi-arc ion plating	316L SS plates	In vitro (in artificial saliva)	The surface roughness was 10.4 nm after being coated.	The Ti-DLC film showed the lowest corrosion current density (~4.577 μA/cm^2^) and thickness reduction (~0.12 μm) in different electrolytes compared to TiN and TiCN.	Wear resistance
[31]	TiN	Direct current reactive magnetron sputtering	316 L SS and (100)-oriented Si substrates	In vitro (in artificial saliva)	NA	The corrosion resistance of TiN decreased with the nitrogen content and carbon content used in the preparation process.	Friction reduction effectiveness Wear resistance
[58]	TiN	Ion plating	SS and NiTi archwires	In vitro (friction test: NA; corrosion test: in 0.9% NaCl solution)	R_a_ of TiN-coated SS archwire increased from 0.023 to 0.046 µm. R_a_ of TiN-coated NiTi archwire remained at 0.001 µm.	Corrosion resistance increased in coated archwires. The pitting corrosion was reduced compared to uncoated archwires. The breakdown potential of the non-coated and the TiN-coated SS wire was 0.46 and 0.61 V (*p* < 0.05), and that of the non-coated and the TiN-coated Ni-Ti wire was 1.20 V and more than 2.0 V, respectively.	Friction reduction effectiveness
[166]	TiN	Multi-arc ion plating	316L SS plates	In vitro (in artificial saliva)	NA	The TiN film showed a corrosion current density of ~8.5 μA/cm^2^, which showed lower corrosion resistance than TiCN and Ti-DLC.	NA
[19]	TiN	Hollow cathode discharge	SS and NiTi archwires	In vitro (in physiological saline, sterile water, 35% hydrochloric acid, and 88% lactic acid)	When SS archwires were coated, R_a_ increased from ~0.025 to ~0.04 μm; when NiTi archwires were coated, R_a_ increased from ~0.14 to ~0.15 μm.	The acid-mediated corrosion and the elution of Ni ions from the wire surface reduced from 2.38, 3.37, 215, 499.84, and 2.29 μg/L to 1.35, 4.16, 13536.28, and 1.44 μg/L after being immersed for 30 min in sterile water, physiological saline, 35% hydrochloric acid, and 88% lactic acid, respectively.	NA
[33]	TiN/Ti	High-power magnetron sputtering deposition	Ti_6_Al_4_V plate	In vitro	The average grain size was 9.1–11.6 μm after being coated.	The 1-layer, 2-layer, and 4-layer TiN/Ti multilayer coatings had similar wear rates which were less than 1/20 of the wear rate of the Ti6Al4V substrate, while the wear rates of the 8-layer and 12-layer TiN/Ti multilayer coatings were higher than that of the Ti6Al4V substrate.	Wear resistant (1-layer, 2-layer, and 4-layer TiN/Ti multilayer coatings)
[177]	Ti-Nb	Laser cladding	Cold-rolled NiTi alloy	In vitro (in simulated body fluid)	NA	I_corr_ reduced from 272.4 to 163.7 nA/cm^2^ and E_corr_ increased from −0.184 to −0.128 V, which exhibited a good corrosion resistance.	NA
[173]	TiO_2_	Sol-gel dip-coating	SS archwires	In vitro (in Ringer’s solution)	NA	I_corr_ increased from 0.007 to 39.9 μA/cm^2^ and E_corr_ reduced from −162 to −300 mV, which exhibited an unsatisfactory corrosion resistance.	NA
[158]	TiO_2_ (NPs)	Atomic layer deposition	316L SS	In vitro (in simulation body fluid solution)	NA	The corrosion rate calculated by current density was 1.913 mpy for an uncoated sample and reduced after coated to 0.203 mpy for titania film, 0.174 mpy for alumina film, and 0.164 mpy for multilayer. The corrosion resistance was effectively enhanced by thin films, multilayer proved to be more corrosion protection than single layers, and Al2O3 had better corrosion resistance than TiO_2_.	Antibacterial effectiveness Biocompatibility
[173]	TiO_2_/Ag	Sol-gel dip-coating	SS archwires	In vitro (in Ringer’s solution)	NA	I_corr_ increased from 0.007 to 30.0 μA/cm^2^ and E_corr_ reduced from −162 to −285 mV, which exhibited an unsatisfactory corrosion resistance.	Wear resistance
[159]	ZrO_2_	EBPVD	316L SS	In vitro (in artificial saliva and artificial saliva containing 0.2% or 2% of NaF)	R_a_ reduced from 10 to 3 nm after being coated.	I_corr_ reduced from 0.34–9.27 to 0.04–0.46 μA/cm^2^; E_corr_ increased from −0.08775 to −0.0739, from −0.2122 to −0.1267, and from −0.2831 to −0.1714 mV when immersed in artificial saliva, artificial saliva containing 0.2% NaF, and artificial saliva containing 2% of NaF, respectively, which exhibited a good corrosion resistance.	Antibacterial effectiveness Wear resistance Biocompatibility

NA: not applicable; E_corr_: corrosion potential values; I_corr_: corrosion current density; aC:H: hydrogenated forms of diamond-like carbon; Cr: chromium; Fe: ferrum; B-HAp: boron-doped hydroxyapatite; NaCl: sodium chloride; GB-PP: grid-blasted plasma-polymerized; AISI: American Iron and Steel Institute; SPM: scanning probe microscope; Ni-Cr: nickel-chromium; Ti-Cr-N: titanium-chromium-nitrogen; RMS: root mean square; TiN/Ti: titanium nitride/titanium; Ti_6_Al_4_V: titanium-aluminum-vanadium alloy with 6% aluminum and 4% vanadium; Ti-Nb: titanium-niobium; TiO_2_/Ag: titanium dioxide/silver.

## 5. Discussion

It is necessary to modify the surface of orthodontic accessories—especially brackets and archwires—using surface coatings. This article reviews previous surface coatings and modification effects related to friction reduction, antibacterial properties, and corrosion resistance.

From the perspective of a research mindset, there are numerous types of functional coatings for the surface of orthodontic attachments. They have different methods and conditions of coating, but all have certain functions under different experimental conditions. This may seem like another new kind of coating has been discovered to choose from; however, in fact, it makes little sense to keep discovering new materials without comparing a range of materials and coating methods horizontally under the same conditions. Therefore, a detailed comparison of newly discovered coating materials with existing materials should be performed in various aspects in order to tailor the most appropriate coating materials for clinical applications. In addition, different coating methods and conditions have different effects on the properties and effects of the coating. When the coating temperature, duration, and/or raw material ratio are changed, the nature of the coating also changes [133]. With this in mind, the best conditions for preparing a certain functional coating can be found by changing the coating process. Moreover, there have been few relevant animal experiments and clinical trials, making it challenging to determine the clinical value of coatings. The use of animals for validation of the effectiveness and safety of various surface coatings is essential, and clinical trials should also be conducted to document what is really happening in clinical applications in more detail.

In clinical practice, it is expected that a single coating on the surface of a fixed orthodontic appliance will meet all of the performance requirements for friction reduction, antibacterial activity, and corrosion resistance. However, it is difficult to find a single coating that can satisfy all three conditions at the same time. A few coatings, such as TaAgB and PTFE, offer relatively comprehensive performance improvements [68,92,97,147,192]. However, studies on these materials are scarce, and there is not enough clear and stable clinical evidence. Recently, new coating materials such as tantalum nitride-copper (TaCuN) solid solution [195] have also been verified with excellent modification properties, in addition to novel coatings in other field, such as ferrum-aluminum/dizinc magnesium (Fe-Al/MgZn_2_) multilayer [196] and nanographene piece + carbon dots and nickel-tungsten (Ni-W) co-deposition coatings [197], but they have not been tested on brackets or archwires. Moreover, the durability of the coatings needs to be further improved. As a long-term fixed intraoral orthodontic accessory, the brackets need to maintain their surface properties for months or more than a year, and the coatings are likely to be damaged and peeled off by sliding [62]. This suggests that functional coatings need to be able to remain effective in the mouth for a long period, in addition to the abovementioned properties. Therefore, coatings with high durability are needed.

The antifriction effect of GSEC and epoxy can last up to 30 days [90,97]. For antibacterial coatings, N-doped TiO_2_ inhibited *Streptococcus mutans* for up to 90 days in vitro and up to 60 days in clinical trials [132,134], making it the most durable material in the scope of this review. Ag and ZnO have also been validated for a period of 3 months [112]. TiO_2_ nanoparticles were effective within 3 weeks [128], and lysosome coating lasted 2 weeks [154]. For titanium sputter coating, corrosion can be reduced within 30 days [175]. Likewise, DLC will have this effect for 14 days [76]. Most of the remaining materials have not been confirmed by long-term experiments and cannot be used in the comparison. Therefore, sustainability aspects should be an important research direction in the future, with considerable significance for the life and validity of the coating.

In a comprehensive view, it is necessary to formulate and compare various surface coating materials and technologies in order to obtain surface coatings that combine multiple functions, are proven to be safe and effective in clinical trials, and have good durability to increase efficiency and reduce risks in the orthodontic treatment process. Among all functional coatings, the best friction-reducing coatings are TiN, TaAgB, and GO, which not only have remarkably low CoF, but also have advantages such as antibacterial properties, wear resistance, corrosion resistance, and good biocompatibility [30,32,58,68,89,181]. The next choices are TiCN, GSEC, CN_x_, and ZnO, which do not have as many additional advantages, but also have excellent friction reduction properties [27,29,31,32,46,90,112]. However, CrN [30], epoxy [93], and parylene [97] are least recommended as friction-reducing coatings because they have been shown in recent studies not to reduce surface friction and sometimes even increase it. Among the antimicrobial coatings, AgNP/PTFE coatings [157], CTS/PEG coatings [156], and F-O-Ti coatings [136] are the most recommended because they have the advantages of corrosion resistance, durability, and biocompatibility, on the basis of antimicrobial rates of over 90%. Ag and ZnO may not be as strong as the former coatings in terms of wear resistance, aesthetics, and durability. However, Ag and ZnO, as classic antimicrobial coatings, have also been repeatedly proven to have excellent antimicrobial properties and biocompatibility, and have been put into clinical application for a longer time and with good results [109,139]. Interestingly, Al compound and composite coatings generally perform better in corrosion-resistant coatings, such as Al_2_O_3_ coatings [30,158], Al_2_O_3_/TiO_2_ multi-layer coatings [158], and Al-SiO_2_ coatings [70]. In particular, Al-SiO_2_ coatings not only significantly reduce the corrosion on the surface of substrates, but also have the advantages of friction reduction, wear resistance, and biocompatibility [70]. This kind of multilayer and composite material is also gradually becoming a research trend in recent years, which might be suitable for application.

The materials and methods already available are adequate, but there are still directions that need to be studied. Some future research prospects are presented here. First, the control of the thickness of the coatings: on the basis of ensuring the modification effect, the thickness of the coating should be reduced as much as possible, which not only saves material but also lessens the adverse effect of the coating’s thickness on the friction [60,78]. Second, it is necessary to verify the clinical effect that the coating can eventually bring—for example, the improvement of the friction-reducing coatings in terms of practical orthodontic efficiency, the contribution of the antimicrobial coatings to avoiding the demineralization of the tooth surface and the inflammation of the periodontal tissues, and the ability of the anticorrosion coatings to reduce patients’ metal-ion allergies. However, the biggest problem with animal experiments is the difficulty of controlling the experimental conditions [198]. Therefore, if the safety of the materials can be confirmed, preclinical trials can be considered directly, without animal experiments. Finally, the long-term effects of various coatings are uncertain. For orthodontic treatment, they can be as short as a few months or as long as a few years. Orthodontic appliances—especially brackets—are present in the oral environment from start to finish, so the coatings on the bracket surfaces need to maintain the longest-lasting modified effect. In fact, laser cladding may offer greater advantages than chemical, electrochemical, and physical deposition methods in terms of coating thickness and permanence. These processes have little effect on the dimensions of the substrates and are less susceptible to peeling off [199]. In this regard, more efforts could be made in laser cladding as a coating method in the future. There are still some limitations to this review. Our discussion in this case is directed only to the surface coating of orthodontic fixed appliances, while other surface modification methods have not been discussed extensively. In the case of orthodontic appliances, the main focus is only on the coating of the two main components—brackets and archwires. Other components of orthodontic appliances, such as ligature wires, buccal tubes, and orthodontic band strips, have not been discussed in depth. Of course, functional surface coatings for orthodontic bands have recently emerged, such as AgNPs coatings with enhanced antimicrobial properties [200]. Going forward, each attachment for fixed appliances should be taken into account for future improvements.

## 6. Conclusions

In order to increase treatment efficiency and reduce side effects, functional surface coatings of orthodontic fixed appliances are receiving increasingly more attention. Over the past decade, research on friction-reducing, antibacterial, and corrosion-resistant coatings has grown, better materials have been discovered, and methods and conditions have been explored to optimize the performance of the materials. However, the current coating materials have not yet achieved the perfect integration of these three functions, and most of the research is limited to in vitro experiments. Thus, there is still a certain distance to go before clinical applications, so further research is needed in the future.

## Figures and Tables

**Figure 1 ijms-24-06919-f001:**
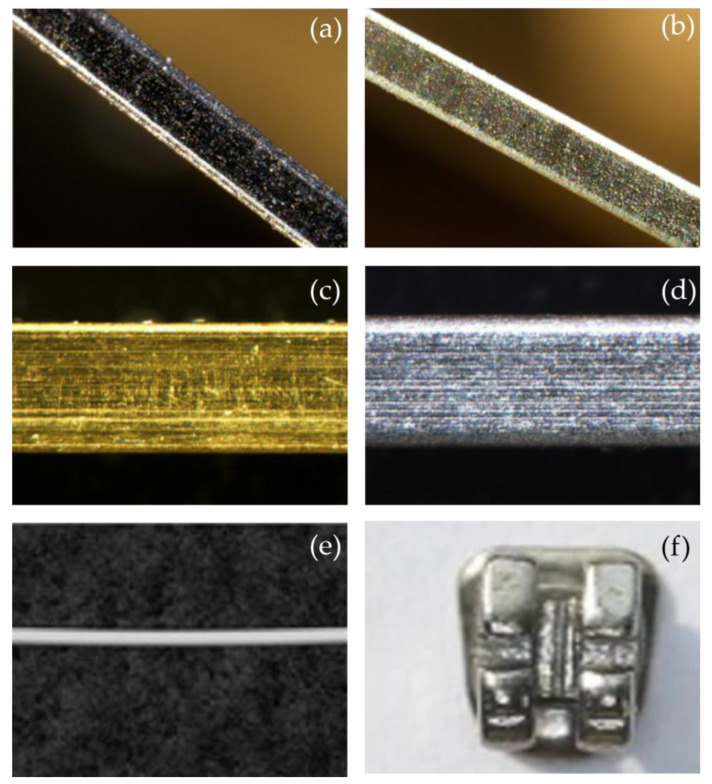
Functional coatings that have been applicated on brackets and archwires. (**a**) nickel (Ni) + molybdenum disulfide (MoS_2_)-coated stainless steel (SS) archwire; (**b**) Ni + tungsten disulfide (WS_2_)-coated SS archwire; (**c**) titanium nitride (TiN)-coated SS archwire; (**d**) TiN-coated nickel-titanium (NiTi) archwire; (**e**) polymer-coated wires (microcoated stainless steel wire^®^, G&H Wire Company, Franklin, IN, USA); (**f**) polydopamine and blue fluorescent hollow carbon dots (PDA-HCD) brackets. Adapted with permission from [18,19,20,21].

**Figure 2 ijms-24-06919-f002:**
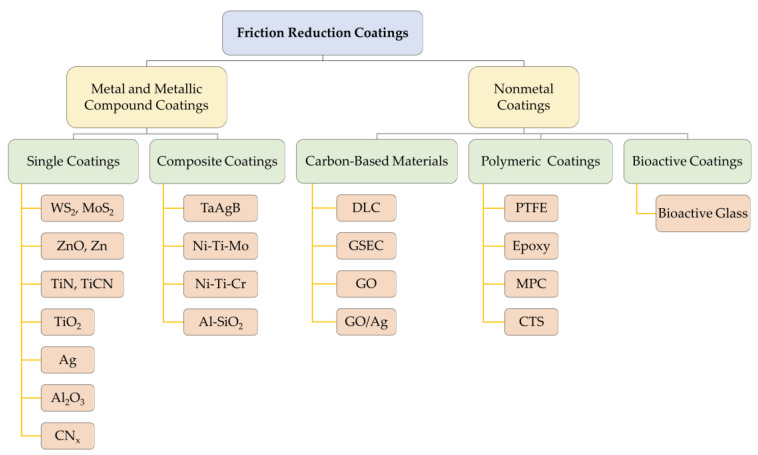
Summary of friction reduction coating materials. WS_2_: tungsten disulfide; MoS_2_: molybdenum disulfide; ZnO: zinc oxide; Zn: zinc; TiN: titanium nitride; TiCN: titanium carbonitride; TiO_2_: titanium dioxide; Ag: silver; Al_2_O_3_: aluminum oxide; CN_x_: carbon nitride; TaAgB: silver-doped tantalum boride; Ni-Ti-Mo: nickel-titanium-molybdenum; Ni-Ti-Cr: nickel-titanium-chromium; Al-SiO_2_: aluminum-silicon dioxide; DLC: diamond-like carbon; GSEC: graphene-sheet-embedded carbon; GO: graphene oxide; GO/Ag: graphene oxide/silver; PTFE: polytetrafluoroethylene; MPC: 2-methacryloyloxyethyl phosphorylcholine; CTS: chitosan.

**Figure 3 ijms-24-06919-f003:**
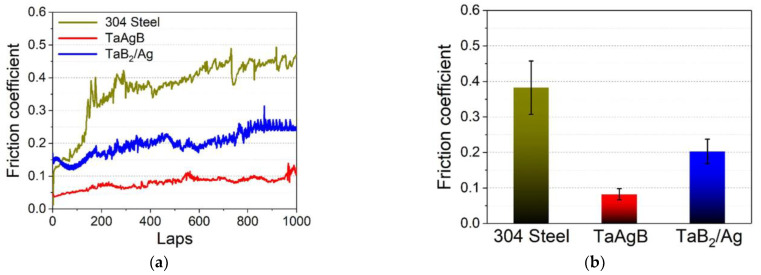
(**a**) Dynamic friction curves for TaAgB coating, TaB_2_/Ag coating, and uncoated 304 steel. (**b**) Mean friction coefficient for TaAgB coating, TaB_2_/Ag coating, and uncoated 304 steel. TaB_2_: tantalum boride. Adapted with permission from [68]. Copyright 2022, Elsevier.

**Figure 4 ijms-24-06919-f004:**
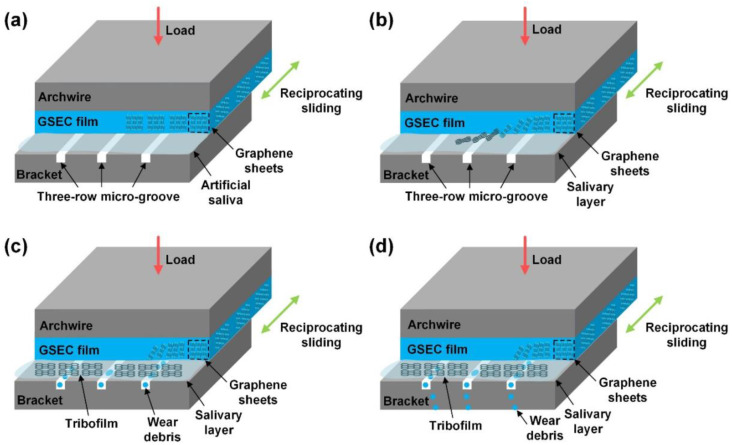
The low-friction and low-wear mechanisms of the GSEC-film-coated stainless steel archwires running against three-row microgroove-textured stainless steel brackets in an artificial saliva environment. (**a**) Initial state of the contact combination, (**b**) stable low friction with the formation of graphene-rich tribofilm and salivary adsorbed layer, (**c**) accumulation of wear debris detached from the GSEC film with microgroove, and (**d**) flow out of wear debris with artificial saliva from the microgroove. Adapted with permission from [46].

**Figure 5 ijms-24-06919-f005:**
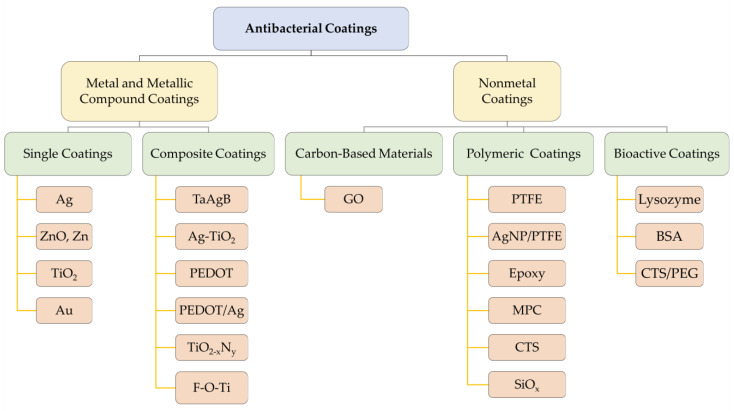
Summary of antibacterial coating materials. Au: gold; Ag-TiO_2_: silver-titanium dioxide; PEDOT: poly(3,4-ethylenedioxythiophene); PEDOT/Ag: poly(3,4-ethylenedioxythiophene)/silver; TiO_2−x_N_y_: nitrogen-doped titanium dioxide; F-O-Ti: fluorine and oxygen double-deposited titanium; AgNP/PTFE: nanosilver/polytetrafluoroethylene; SiO_x_: silicon oxides; BSA: bovine serum albumin; CTS/PEG: chitosan/polyethylene glycol.

**Figure 6 ijms-24-06919-f006:**
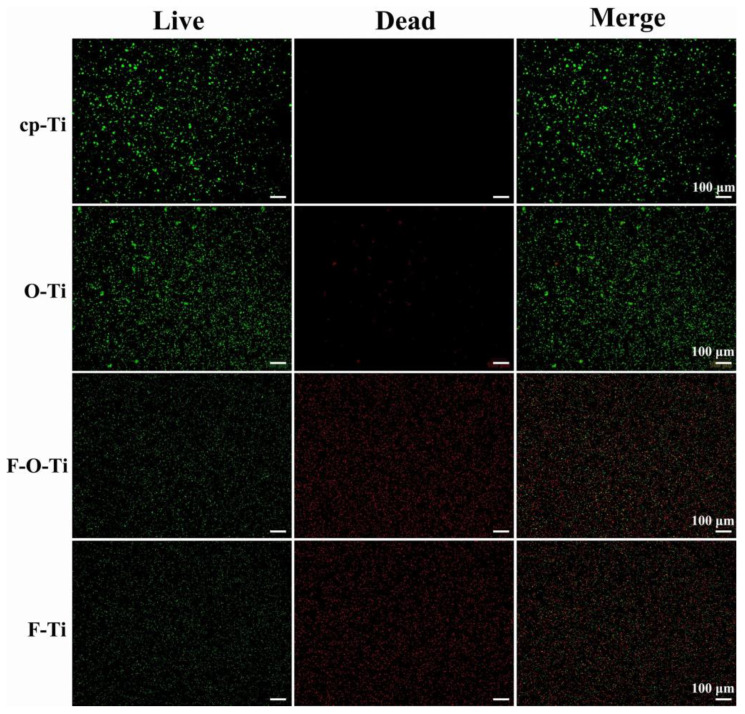
Live (green) and dead (red) staining of *Staphylococcus aureus* on different samples after culturing for 1 day. Adapted with permission from [136]. Copyright 2022, Elsevier.

**Figure 7 ijms-24-06919-f007:**
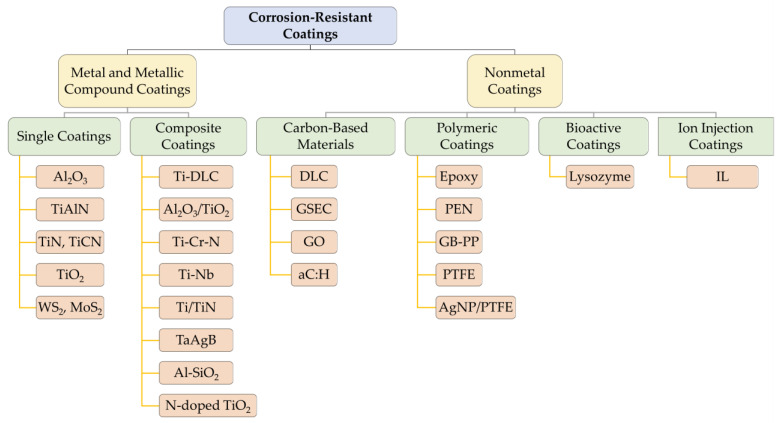
Summary of antibacterial coating materials. TiAlN: titanium aluminum nitride; Ti-DLC: titanium-diamond-like carbon; Al_2_O_3_/TiO_2_: aluminum oxide/titanium dioxide; Ti-Cr-N: titanium-chromium-nitrogen; Ti-Nb: titanium-niobium; Ti/TiN: titanium/titanium nitride; N: nitrogen; aC:H: hydrogenated forms of diamond-like carbon; PEN: polyethylene naphthalate; GB-PP: grid-blasted plasma-polymerized; IL: ionic liquid.

**Figure 8 ijms-24-06919-f008:**
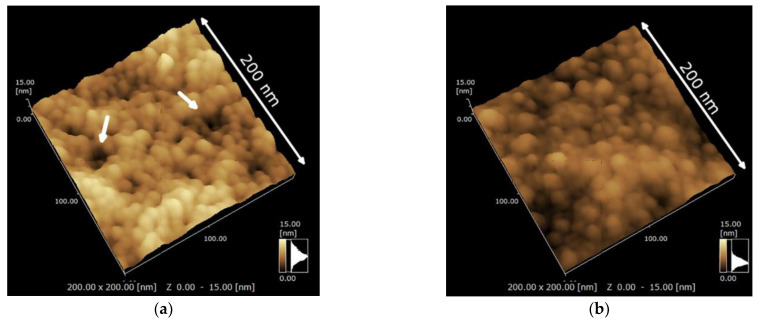
Scanning probe microscope (SPM) images of (**a**) current-anodized and (**b**) pulse-anodized NiTi surfaces. The white arrows show the pores on the surfaces of the coating. Adapted with permission from [174]. Copyright 2022, Elsevier.

**Figure 9 ijms-24-06919-f009:**
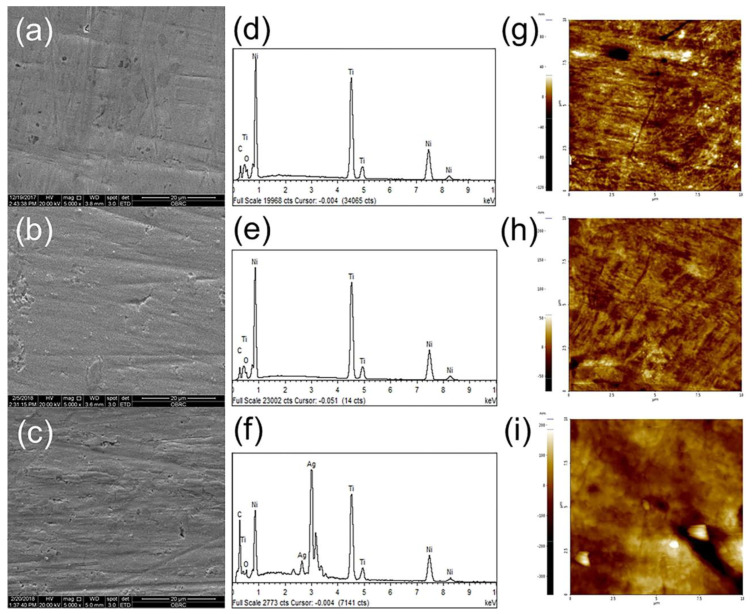
(**a**–**c**) Scanning electron microscope (SEM) images, (**d**–**f**) energy-dispersive spectroscopy (EDS) analysis, and (**g**–**i**) atomic force microscope (AFM) images of the bare NiTi, GO-coated NiTi, and GO/Ag-coated NiTi alloy, respectively. Adapted with permission from [181].

## Data Availability

Not applicable.

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
