# Peer review of "Functional Surface Coatings on Orthodontic Appliances: Reviews of Friction Reduction, Antibacterial Properties, and Corrosion Resistance"

_ijms, 2023, doi:10.3390/ijms24086919_

Round 1

Reviewer 1 Report

In the manuscript “Functional Surface Coatings on Orthodontic Appliances: Reviews of Friction Reduction, Antibacterial and Corrosion Resistance”, Zhang et al. review functional surface coatings applied to the orthodontic field. The review needs to be major revised to address issues. The suggestions are divided into categories.

CITATIONS. There are many citations missing. Here just a few have been pointed out, please double-check the entirety of the manuscript.

·        Line 71. Add citation to the statement related to “tungsten disulfide”.

·        Lines 71-73. Same issue, lack of citations.

·        Line 76. No citation regarding the techniques.

·        Lines 103-113. No citations at all.

·        Lines 135-147. No citations at all.

·        Line 192. Please add citation to the statement.

·        Line 195-199. Add citation.

GENERAL.

·        Lines 57-59. Please expand the issue of the release of metal-ions.

·        Lines 128-131. Please update using appropriate academic English.

·        Line 272-274. What is the reason for the different behavior in friction reduction?

·        Lines 501-503. Could you expand the discussion on why Ag and ZnO are better than GO?

·        Lines 509-513. The usage of silver as an antibacterial coating needs to be better explained. What is the mechanism behind it? Suggestion:

o   https://doi.org/10.2147/IJN.S246764

o   https://doi.org/10.3389/fmicb.2016.01831

The authors focus on the usage of functional surface coatings only in the orthodontic field, which is the goal of the review. However, insights from other fields could improve the review. A few suggestions are made below.

·        Lines 101-102. The usage of MoS2 and WS2 coatings under wet conditions has been tested in different applications. For instance, in 2D semiconductors:

o   https://doi.org/10.1021/acs.langmuir.5b02057

o   https://doi.org/10.1021/nn5072073

·        The section about antibacterial coating has focused on materials that rely on antibacterial based on the chemistry of the material. However, many antibacterial surfaces can reduce attachment based on surface topography by controlling roughness. The review covers a few studies but the mechanism behind the antibacterial property based on roughness is not discussed. Despite not being totally related to orthodontic, as the mechanisms are the same, suggested papers to enhance the discussion:

o   10.1088/1748-3190/ac060f

o   10.1021/acsami.1c22205

·        The discussion section could be improved with more insights, from the authors, for possible future research.

Reviewer 2 Report

Dear authors, 

This review is interesting however some mistakes should be solved:

Keywords should be organized according to mesh terms and alphabetical order.

The use of full name before abbreviation is a must.

Please add full name for the table in the caption.

Add limitations for the study.

Reviewer 3 Report

reviewer comments on "Functional Surface Coatings on Orthodontic Appliances: 2 Reviews of Friction Reduction, Antibacterial and Corrosion 3 Resistance"

The review is very comprehensive, literature review tables are provided and well organized. Following are some points the authors can add to improve

1. Add future aspects heading and include 2023 papers that are very novel in this field. Give future perspectives

2. Add sustainability aspects as well in the discussion chapter. Lifecycle assessment. of various coatings and how well they suit. 

3. There is no quantitative data. For instance Table 3 corrosion, there is no comparison with the base material, dental etc. to show a percentage improvement in corrosion. Either make a figure on how much performance improvement is given or add it to the table. For example, the authors added, "Ti-Cr-N coatings can reduce SS304's release of chromium species by about 67%,". So this is quantitative and so more of this should be visible in the table or figure. 

4. Similarly add friction coefficient values or put them as figures with references for table 1 as well. Table 1 also adds the dominant wear mechanism column and how much times or percentage wear resistance has improved as another column. 

5. Microstructure or cross-sectional surface and surface roughness information is not available. Which type of coating produces what roughness? 

6. In the discussion section 5 you need to rank the coatings based on Friction, corrosion, biocompatibility, and wear resistance. and give you input as to which is more suitable. 

7. Resent patents have not been included and the applications of these coatings that are commercially available should be also provided. Give 6 or 7 figures of the recent adoption of coatings technology for dental application as a cluster of figures that target only applications. 

8. Cross section of each type of coating should be provided from the latest papers already cited in the manuscript. 

9. The adhesion strength values are not given for different values. 

Reviewer 4 Report

This review article reports on the developments and testing of functional coatings for orthodontic appliances. Description of coatings from different classes ranging from metals and compounds to polymers and other carbon based coatings and their performance regarding friction reduction, bacterial and corrosion resistance are provided. The article is informative on very relevant topics, and gives critical comments on the current state of development of such coatings, and therefore is suited for publication after some points are revised:

- There is a confusion in the classification of deposition techniques. For instance, in the abstract the autors state: "In addition to single use, metal-metal or metal-nonmetal materials can be combined. Methods of coating preparation include, but are not limited to, direct current sputtering, physical vapor deposition (PVD), radio frequency magnetron sputtering, and so forth, with a variety of different conditions for preparing the coatings.", however sputtering (RF and DC) are a type of Physical Vapor Deposition (PVD), so please verify your classification criteria of the techniques along the text.

- The first paragraph of introduction should be entirely revised as there are repeated phrases.

- The classification of oxides and nitride coatings such as ZnO, Al2O3 or TiN as metallic films, in the same category of Ag, is not correct in my opinion. I suggest calling them metallic compound coatings or simply Oxides or Nitrides.

- In line 245 it is said that "The method of using low-pressure metal-organic chemical vapor deposition (LPMOCVD) can control the film thickness by adjusting the deposition time[48], which is a viable approach.", but controlling the film thickness by changing the deposition time is feasible in several deposition techniques. I do not agree that it should be stated as a differential feature of LPMOCVD.

- In the caption of Figure 2, please be more specific about the materials you are describing rather than calling it "another coating".

- In figure 3 it is not clear what are the differences between panels a) to d). Please, either describe the mechanisms illustrated or remove the figure.

- In tables 1, 2 and 3 the "Coating Methods" columns should be entirely revised. In some cases the name of the technique is provied, but in others there are other information that does not allow the reader to understand which technique was employed. Ex: "InVu Roth, (TP Orthodontics, LaPorte, Ind., USA), System Al-exander LTS (AO. American Or-thodontics, Wisconsin., EE. UU), Gemini Roth (3 M Unitek, Mon-rovia, CA., USA), Nu-Edge Roth (TP Orthodontics, LaPorte, IN, USA), Radiance plus Roth (AO. American Orthodontics, Wis-consin., EE. UU)"

Round 2

Reviewer 1 Report

The authors have made necessary revisions to the revised manuscript. 

Reviewer 3 Report

Reviewer comments on "Functional Surface Coatings on Orthodontic Appliances: Reviews of Friction Reduction, Antibacterial and Corrosion Resistance", submitted as ijms-2215639 revised version. 

The authors have made necessary revisions to the revised manuscript.